# Varicella zoster virus encodes a viral decoy RHIM to inhibit cell death

Megan Steain[1]☯*, Max O. D. G. Baker[2]☯, Chi L. L. Pham[2], Nirukshan Shanmugam[2], Yann Gambin[3], Emma Sierecki[3], Brian P. McSharry[1,4], Selmir Avdic[1], Barry Slobedman[1], Margaret Sunde[2‡], Allison Abendroth[1‡*]

**1** Discipline of Infectious Diseases and Immunology, School of Medical Sciences, The University of Sydney, NSW, Australia, **2** Discipline of Pharmacology, School of Medical Sciences and Sydney Nano Institute, The University of Sydney, NSW, Australia, **3** EMBL Australia Node in Single Molecule Sciences, School of Medical Science, University of New South Wales, Sydney, NSW, Australia, **4** School of Microbiology & APC Microbiome Ireland, University College Cork, Cork, Ireland

☯ These authors contributed equally to this work.
‡ MSu and AA also contributed equally to this work.
* megan.steain@sydney.edu.au (MS); allison.abendroth@sydney.edu.au (AA)

## Abstract

Herpesviruses are known to encode a number of inhibitors of host cell death, including RIP Homotypic Interaction Motif (RHIM)-containing proteins. Varicella zoster virus (VZV) is a member of the alphaherpesvirus subfamily and is responsible for causing chickenpox and shingles. We have identified a novel viral RHIM in the VZV capsid triplex protein, open reading frame (ORF) 20, that acts as a host cell death inhibitor. Like the human cellular RHIMs in RIPK1 and RIPK3 that stabilise the necrosome in TNF-induced necroptosis, and the viral RHIM in M45 from murine cytomegalovirus that inhibits cell death, the ORF20 RHIM is capable of forming fibrillar functional amyloid complexes. Notably, the ORF20 RHIM forms hybrid amyloid complexes with human ZBP1, a cytoplasmic sensor of viral nucleic acid. Although VZV can inhibit TNF-induced necroptosis, the ORF20 RHIM does not appear to be responsible for this inhibition. In contrast, the ZBP1 pathway is identified as important for VZV infection. Mutation of the ORF20 RHIM renders the virus incapable of efficient spread in ZBP1-expressing HT-29 cells, an effect which can be reversed by the inhibition of caspases. Therefore we conclude that the VZV ORF20 RHIM is important for preventing ZBP1-driven apoptosis during VZV infection, and propose that it mediates this effect by sequestering ZBP1 into decoy amyloid assemblies.

## Author summary

RIP homotypic interaction motifs (RHIMs) are found in host proteins that can signal for programmed cell death and in viral proteins that can prevent it. Complexes stabilized by intermolecular interactions involving RHIMs have a fibrillar amyloid structure. We have identified a novel RHIM within the ORF20 protein expressed by Varicella zoster virus (VZV) that forms amyloid-based complexes with human cellular RHIMs. Whereas other herpesvirus RHIMs inhibit necroptosis, this new VZV RHIM targets the host RHIM-

**Data Availability Statement:** All relevant data are within the manuscript and its Supporting Information files.

**Funding:** This work was supported by funding from the Australian Research Council to M. Sunde

and E.S. (DP180101275), Research Training Program support to MB, a BioMed Connect Grant from the Sydney Medical School to M. Sunde and A. A., a University of Sydney Bridging Scheme grant to M. Steain and National Health and Medical Research Council funding to A. A. The funders had no role in study design, data collection and analysis, decision to publish, or preparation of the manuscript.

**Competing interests:** The authors have declared that no competing interests exist.

containing protein ZBP1 to inhibit apoptosis during infection. This is the first study to demonstrate the importance of the ZBP1 pathway in VZV infection and to identify the role of a viral RHIM in apoptosis inhibition. It broadens our understanding of host defense pathways and demonstrates how a decoy amyloid strategy is employed by pathogens to circumvent the host response.

## Introduction

Viruses have evolved a diverse range of strategies to evade host intrinsic, innate and adaptive immune responses. Members of the *Herpesviridae* family, Herpes simplex virus (HSV) -1 and human and murine cytomegalovirus (HCMV/MCMV), are masters at manipulating host cell death pathways such as apoptosis and necroptosis, in order to successfully spread and establish latency [1–3]. Although Varicella zoster virus (VZV) causes a significant health burden [4–6], the mechanisms employed by VZV to undermine host responses have not been fully elucidated. Primary infection with VZV leads to varicella, commonly known as chickenpox. During this infection the virus establishes latency within sensory neurons, and when VZV-specific T cell immunity wanes, the virus can reactivate to result in herpes zoster (shingles) [7]. Complications arising from VZV reactivation include protracted pain termed post-herpetic neuralgia, encephalitis and VZV vasculopathy (reviewed in [8]).

VZV is a highly cell-associated virus and does not release cell-free virions into culture [9], necessitating cell-associated propagation of the virus *in vitro*. VZV is also highly species-specific, not progressing to productive infection in cells of non-human origin [10]. Therefore no animal model exists which recapitulates the full spectrum of VZV disease. Combined, these properties of VZV have largely hampered the characterization of VZV pathogenesis.

Cell death can be initiated upon sensing of a viral pathogen as a means of prematurely aborting the virus lifecycle and thereby limiting the spread of infection. Apoptosis and necroptosis may form part of the antiviral response [2, 11–14]. The most well characterized pathway of necroptosis is that induced by tumour necrosis factor (TNF). Upon TNF binding to its receptor (TNF-R1), cellular inhibitor of apoptosis proteins (cIAP 1 and 2) mediate K63-ubiquitination of receptor interacting protein kinase 1 (RIPK1), as part of a signalling platform known as complex I, which drives NF-κB signalling (reviewed in [15]). If cIAPs are inhibited (e.g. by SMAC mimetics), RIPK1 can associate with FADD, caspase 8 and cFLIP (termed complex II). This can lead to the cleavage of caspase 8 to drive apoptosis [16, 17]. However, if caspase 8 is also inhibited then RIPK1 can interact with receptor interacting protein kinase 3 (RIPK3) to drive necroptosis [18, 19].

RIPK1 and RIPK3 interact via highly conserved RIP Homotypic Interaction Motifs (RHIMs), leading to the assembly of the necrosome complex [20]. The RIPK1:RIPK3 necrosome complex has a fibrillar structure and is stabilised by an amyloid cross-β core, formed by interactions between the RHIMs of the two component proteins [21]. Generation of the necrosome results in the phosphorylation of RIPK3, which subsequently phosphorylates mixed lineage kinase domain-like protein (MLKL) [22]. Phosphorylated MLKL drives membrane permeabilisation, resulting in rupture of the cell and necroptotic cell death [23, 24].

The cellular Z-DNA binding protein 1 (ZBP1, also known as DAI) contains at least two RHIMs (A and B) and can also interact with RIPK3 [25–27]. ZBP1 was originally identified as a cytosolic DNA sensor [28], however more recently, studies with Influenza A virus (IAV) and MCMV have shown that ZBP1 can sense viral RNA [29, 30] and simultaneously activate

apoptosis and necroptosis in the same population of cells [29]. This implies that the inhibition of caspase 8 is not always necessary for necroptosis to proceed via this pathway.

Some viruses, including HSV-1, HSV-2, HCMV and MCMV, have evolved mechanisms to inhibit caspase 8 and therefore extrinsic apoptosis [31–33], as well as necroptosis [2, 34, 35]. It has been suggested that necroptosis evolved as a backup mechanism to ensure the elimination of virally infected cells [2]. This theory is supported by studies showing that mice which are deficient in RIPK3, a kinase essential for necroptosis, succumb to fatal vaccinia virus infection whereas wild type mice do not [36].

Many of the currently recognized viral protein inhibitors of necroptosis contain a RHIM [37]. MCMV encodes a RHIM close to the N-terminus of a non-functional ribonucleotide reductase, M45. This virally-encoded RHIM has been shown to interact with RIPK3 to inhibit necroptosis [34, 38]. Furthermore, mutation of MCMV to remove the ability of the virus to inhibit necroptosis renders MCMV highly attenuated *in vivo* [34]. We recently proposed a mechanism by which M45 subverts RHIM-based cell death signalling, by forming heteromeric decoy amyloid structures [39]. We demonstrated that M45, like human RHIM proteins, is able to spontaneously form amyloid fibrils. We also demonstrated that M45 is capable of forming networks of hybrid, heteromeric amyloid structures with RIPK1 and RIPK3, in a manner that is more favourable than the interaction of the two human proteins with each other. It is likely that these human:viral protein complexes, by some property of their conformation, are unable to signal to downstream effectors, and thus cell death signalling is abridged.

HSV-1 and -2 also contain RHIMs in the N-terminal regions of functional ribonucleotide reductases infected cell protein (ICP)6 and ICP10 respectively [40–42]. ICP6 and ICP10 have been shown to block necroptosis in cells of human origin in response to TNF and FasL [41, 42]. Further, ICP6 has been reported to protect human cells from ZBP1-induced cell death [43].

VZV, HSV-1 and HSV-2 all belong to the *Alphaherpesvirinae* subfamily and share a high degree of homology. Thus, we sought to determine if VZV also contained a RHIM that could inhibit necroptosis. We identified a RHIM sequence within the open reading frame (ORF) 20 capsid triplex protein. Like the well-characterised RHIMs in RIPK1, RIPK3 and M45, this RHIM is able to drive the formation of amyloid structures and the ORF20 RHIM interacts with RIPK3 and ZBP1 RHIMs *in vitro*. Strikingly however, whilst VZV can inhibit TNF-induced necroptosis, this effect cannot be attributed to the presence of the RHIM in the VZV ORF20 protein. Instead, this newly identified viral RHIM interacts with ZBP1 and its primary function appears to be the suppression of ZBP1-driven apoptosis through the formation of decoy hybrid amyloid structures.

## Results

### VZV ORF20 contains a RHIM

Given that in HSV-1 and HSV-2 the large subunit of the ribonucleotide reductase contains a functional RHIM [40–42, 44], we examined the amino acid sequence of the orthologue in VZV, which is encoded by ORF19. The C-terminal domains of the ribonucleotide reductase R1 from HSV-1 and VZV (ICP6 and ORF19 respectively) share 44% homology at the amino acid level [45], however we found that VZV ORF19 encodes a shorter R1 that lacks a RHIM. We then examined the remainder of the VZV genome to look for a potential RHIM. We found a RHIM-like sequence within ORF20, which encodes the capsid triplex subunit 1 (Fig 1A and 1B). This putative RHIM is located in the N-terminal region of ORF20, between L24 and Y40 (Fig 1B). ORF20 is homologous to the HSV-1 capsid triplex subunit protein VP19C [46]. There is low conservation in the first 111 amino acids of VP19C compared to orthologue

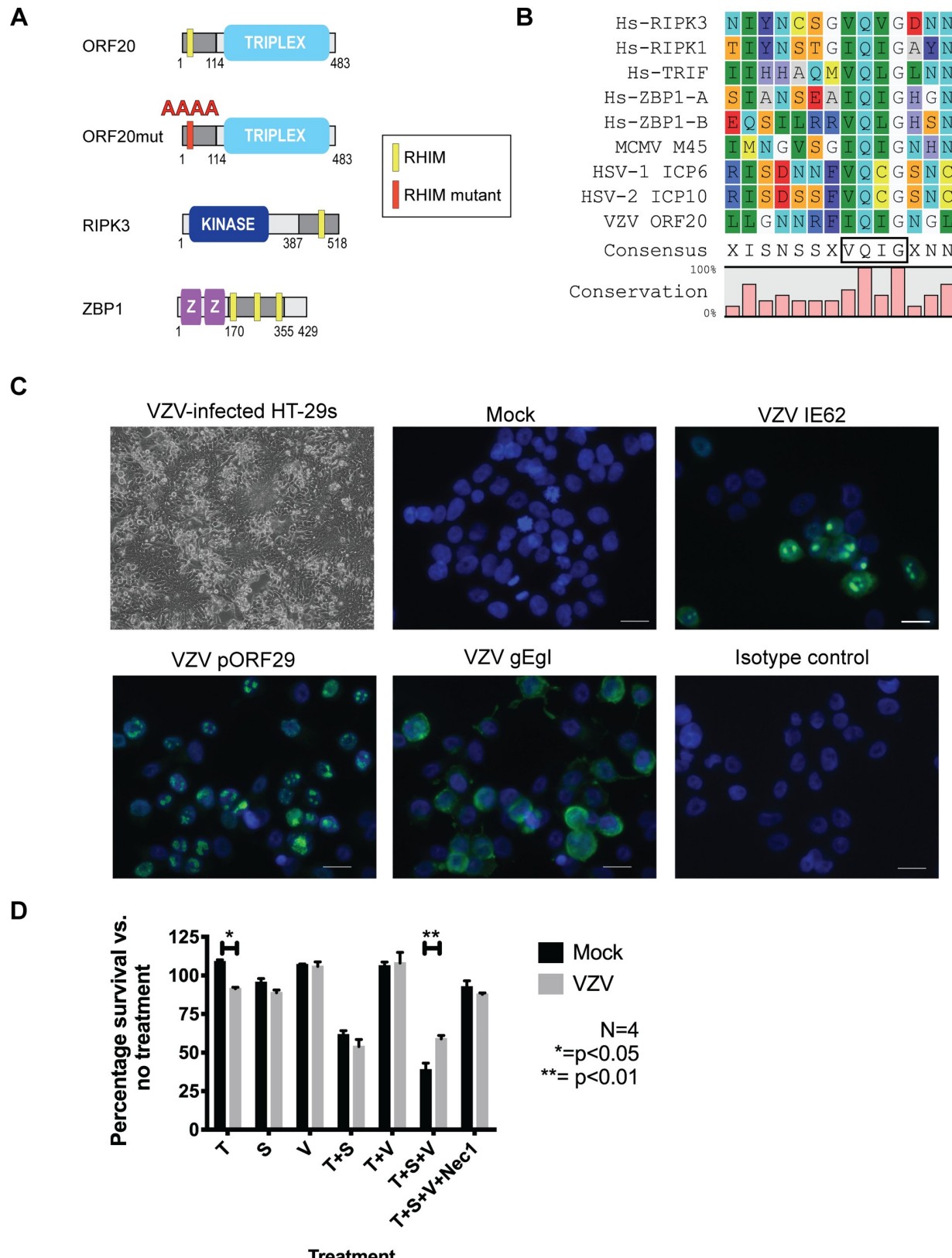

**Fig 1. VZV ORF20 contains a RHIM that inhibits TNF-induced necroptosis in HT-29 adenocarcinoma cells.** (A) Schematic diagram of RHIM-containing proteins. RHIM location indicated by yellow box. (B) Amino acid sequence alignment of the new RHIM identified in VZV ORF20 (Dumas strain) with other

known human cellular RHIMs from RIPK3, RIPK1, TRIF and ZBP1 and the viral RHIMs from MCMV (M45), HSV-1 (ICP-6) and HSV-2 (ICP10), also indicating the percentage of conservation and consensus sequence, with the RHIM core tetrad boxed. (C) Cytopathic effect and immunofluorescence staining for VZV antigens (IE62, pORF29 and the gE:gI glycoprotein complex, green) in VZV-infected and mock HT-29s. Following immunostaining cells were counterstained with DAPI (blue). Scale bars indicate 20 μm (D) Viability of mock and VZV infected HT-29s (72 h post-infection) following treatments with TNF (T; 30 ng/ml), BV-6 (S; 1 μM), z-VAD-fmk (V; 25 μM) and necrostatin-1 (Nec1; 30 μM) alone or in combination as indicated. Data was normalised to DMSO only control. Error bars show standard error of the mean, from 4 independent replicates, statistical significance was determined using a two-way ANOVA.

proteins in other herpesviruses, and mutational analysis has revealed that capsid assembly is not disrupted by alterations to the first 107 codons of VP19C [46]. It is therefore possible that the N-terminal of ORF20 may serve as a cell death inhibitor without disrupting capsid function. The core tetrad of residues in this potential VZV RHIM is IQIG, conforming to the I/V-Q-I/V/L-G consensus found in cellular RHIMs [20], and a number of other residues are conserved. This potential RHIM was conserved within every VZV genome that we examined, including the vaccine strains Varivax and VarilRix (S1A Fig). We then extended our search and found that at least the core tetrad residues of this potential RHIM (I/V-Q-I/V-G) were conserved in other members of the *Varicellovirus* genus for which an ORF20 orthologue sequence was available, including Simian Varicella Virus (SVV), Pseudorabies virus (PRV), bovine herpesvirus (BHV) 1 and 5 and equine herpesvirus (EHV) 1, 4, 8 and 9 (S1B Fig). This level of conservation strongly suggests that this motif is essential for the successful dissemination of *Varicelloviruses*.

## Varicella zoster virus can infect HT-29 adenocarcinoma cells and block TNF-induced necroptosis

Many cultured cells, such as human fibroblasts (HFs), which are commonly used to grow and study VZV infection, lack the expression of RIPK3, rendering them resistant to necroptosis [47]. Further, VZV is highly-species specific and can only establish a full productive infection in cells of human origin [48]. Therefore, in order to study the modulation of necroptosis pathways by VZV we had to identify a cell line that was susceptible to both VZV infection and necroptosis. The human colorectal adenocarcinoma cell line HT-29 has been used extensively to study necroptosis [19, 22, 41, 47], however to date there have been no reports of productive VZV infection in this cell line. In order to determine if VZV could infect HT-29s, cells were grown to ~70% confluence and then VZV rOKA-infected HFs were added at a ratio of 1:8. A cell-associated infection was used due to the highly cell-associated nature of VZV infection *in vitro* [9] and this approach is routinely used to infect cells *in vitro* [49–52]. Within 72 h cytopathic effect (CPE) was readily observed within VZV-infected HT-29s (Fig 1C) and further passaging of the virus in a cell-associated manner in HT-29s could be continued. In addition, following several passages to eliminate the infecting HF inoculum, immunofluorescence staining for VZV immediate early (IE62), early (pORF29) and late proteins (gE:gI complex) was performed. This showed that the full cascade of VZV gene expression occurred in HT-29s, and the cellular localisation of each viral antigen was typical of a productive infection [53–55] (Fig 1C) Together this shows that VZV is able to productively infect HT-29 adenocarcinoma cells.

In order to determine if VZV infection could confer resistance to necroptosis, VZV-infected HT-29s (72 h post-infection, 24–45% gE:gI antigen +) and mock-infected HT-29s were treated with combinations of TNF (T), the Smac mimetic BV-6 (S) and z-VAD-fmk (V) to inhibit caspase 8. The percentage of surviving cells was then determined 17–18 h post-treatment by measuring intracellular ATP levels. On average from four biological replicates, treatment of the cells with TNF alone reduced cell survival in the VZV infected cells compared to mock to a modest yet significant degree, although both mock and VZV-infected HT-29 cells were equally susceptible to apoptotic cell death induced by T+S treatment (Fig 1D). However,

following treatment to induce necroptosis (T+S+V), significantly more cells from the VZV infection survived compared to mock (on average 69% vs. 39%) (Fig 1D). The addition of necrostatin-1 inhibited the cell death induced by T+S+V in both the mock and VZV infected populations, confirming the involvement of RIPK1 in this cell death pathway. Additionally, mock and VZV-infected cells treated with combinations of T+S+V were collected for immunoblot (IB) analysis. These analyses showed that phosphorylated MLKL (pMLKL) a hallmark of necroptosis induction could be readily observed in mock infected, but not in VZV infected cells, treated with T+S+V (S1C Fig). VZV infection also did not appear to alter total cellular levels of MLKL. Together, these results suggest that VZV infection confers a resistance to TNF-induced necroptosis.

## VZV infection inhibits phosphorylation of MLKL

The cell-associated nature of VZV infections means that within the VZV-infected culture there is a mix of bystander and virally infected cells. In order to determine if VZV-infection prevented the phosphorylation of MLKL only within infected cells, dual immunofluorescence staining for VZV antigen and pMLKL was performed. Mock and VZV-infected HT-29s were treated with DMSO alone (control) or with T+S+V for 7–8 h, then fixed and immunostained for the VZV immediate early viral protein IE62 (green) and pMLKL (red). Within the mock-infected HT-29 cultures treated with T+S+V, pMLKL could be readily seen localised to cellular membranes (Fig 2A), with an average of 21.8% (range: 15.9–28.8%) of cells pMLKL positive over three biological replicates (Fig 2B). Within the VZV-infected HT-29 culture, pMLKL staining was observed predominantly in VZV-antigen negative cells following treatment (Fig 2A), with an average of 10.8% (range 8.6–15%) of antigen negative cells being positive for pMLKL and significantly fewer of the VZV-antigen positive cells (on average 3.4%, range 1.1–6.5%), (Fig 2B). No specific pMLKL staining was observed in DMSO control treated cells in the mock or VZV-infected cultures (Fig 2A). This data strongly suggests that VZV infection can confer resistance to TNF-induced necroptosis prior to the phosphorylation of MLKL.

## The VZV ORF20 RHIM does not inhibit TNF-induced necroptosis

In order to determine if the RHIM-like sequence identified within VZV ORF20 was responsible for the observed inhibition of TNF-induced necroptosis, we sought to express ORF20 in isolation in HT-29s. Initially we attempted a lentivirus-based stable-transduction approach and tested three different lentivirus constructs with ORF20 being driven by either a CMV or EF1α promoter. We also tested a transient transduction system using replication deficient adenoviruses. However with both approaches we found we could not achieve a high percentage of HT-29s expressing VZV ORF20, despite each vector driving robust ORF20 expression in 293T cells and/or HFs. Thus, we went on to mutate the ORF20 RHIM within VZV via "recombineering" using the VZV bacterial artificial chromosome (BAC). A tetra-alanine substitution was made to the core amino acids of the ORF20 RHIM (IQIG to AAAA), as this mutation is commonly used to disable RHIM functions (Fig 1A). Previous studies of the viral RHIMs in MCMV and HSV-1 have shown that a tetra-alanine substitution in the RHIM core largely abolishes the ability of these viruses to inhibit necroptosis [34, 41]. Following the production of both the parental virus and the ORF20 RHIM mutated virus (VZV-RHIMmut) in ARPE-19 epithelial cells, the virus was transferred to HT-29s and passaged several times (5+) to ensure there were no carry over ARPEs remaining. Cell survival was then assessed in mock and VZV infected HT-29s following treatments with combinations of T, S and V as outlined above. To ensure that results could be compared between the parent (BAC-derived pOKA) and the mutant virus, the percentage of HT-29s expressing VZV gE:gI glycoprotein complex was

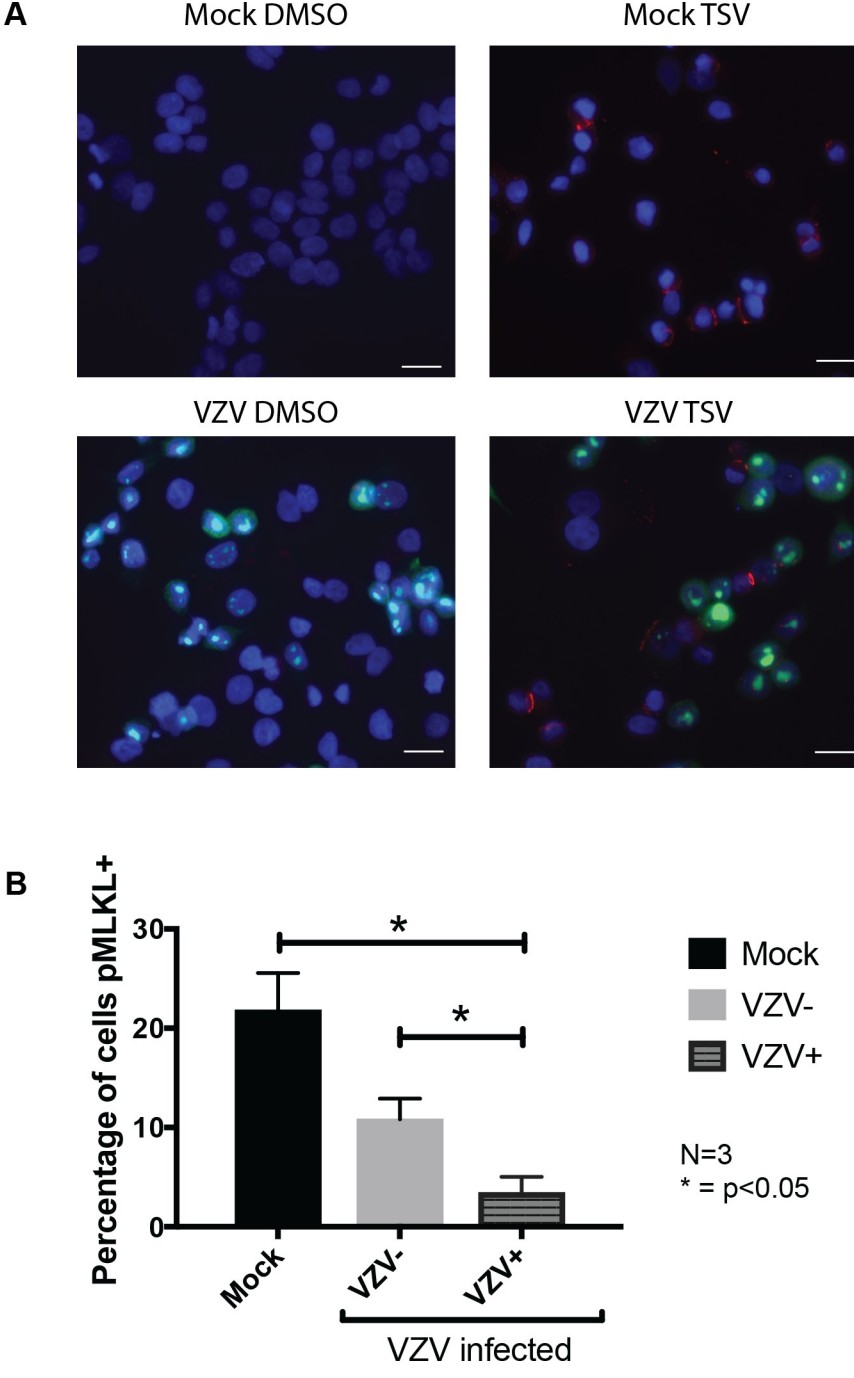

**Fig 2. VZV infection inhibits phosphorylation of MLKL during TNF-induced necroptosis.** (A) Immunofluorescence staining for phosphorylated MLKL (red) and VZV IE62 antigen (green) in mock and VZV infected HT-29 adenocarcinoma cells untreated (DMSO control) or treated with TNF (T; 30 ng/ml), BV-6 (S; 1 μM) and z-VAD-fmk (V; 25 μM) for 7–8 h to induce necroptosis. Following immunostaining cells were counterstained with DAPI (blue). (B) The percentage of cells that were pMLKL positive was determined by randomly imaging 10–20 non-overlapping regions of each slide and manually counting cells from 3 independent experiments. Error bars show standard error of the mean, statistical significance was determined using a one-way ANOVA. Scale bar indicates 20 μm.

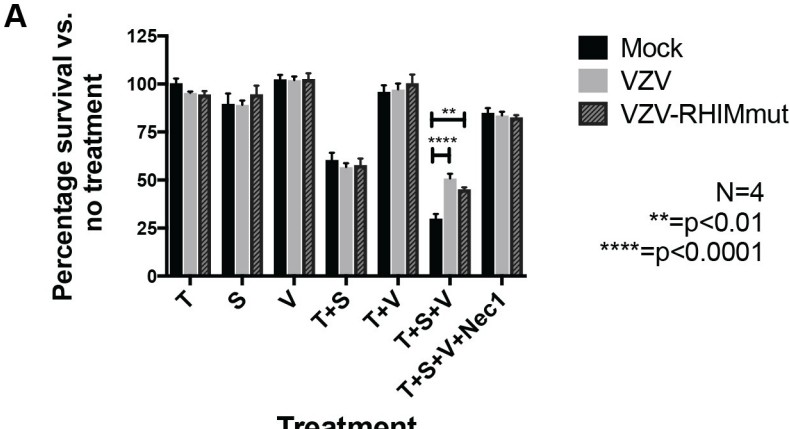

**Fig 3. The VZV ORF20 RHIM is not involved in TNF-induced necroptosis.** (A) Viability of mock, VZV and VZV RHIM mutated (VZV-RHIMmut) virus infected HT-29s (72 h post-infection) following treatments with TNF (T; 30 ng/ml), BV-6 (S; 1 μM), z-VAD-fmk (V; 25 μM) and necrostatin-1 (Nec1; 30 μM) alone or in combination as indicated. Data was normalised to DMSO only control. Error bars show standard error of the mean, from 4 independent replicates and statistical significance was determined using a two-way ANOVA.

determined by flow cytometry at the beginning of each assay for both viruses (S2 Fig). This analysis demonstrated that the percentage of VZV antigen positive cells in the parental and VZV-RHIMmut infected HT-29 cultures was very similar. As seen previously for VZV rOKA (Fig 1), the BAC derived VZV pOKA strain was still capable of inhibiting necroptosis compared to mock (50.7% cell survival versus 29.9%). Surprisingly however, mutated virus with the tetra-alanine substitution of the RHIM (VZV-RHIMmut) protected cells almost to the same extent (45% cell survival) (Fig 3). To further investigate the role of the RHIM, we constructed a virus that had the entire 20 amino acids of the ORF20 RHIM removed (VZV-RHIMKO). This virus was also able to inhibit necroptosis when compared to mock HT-29, however to a slightly lesser extent than what was seen for VZV pOKA (S2B Fig). This result suggests that the VZV ORF20 RHIM does not play a major role in the inhibition of TNF-induced necroptosis in HT-29 cells, and that other viral mechanisms are involved.

## The VZV ORF20 RHIM inhibits ZBP1-driven apoptosis during infection

Pathways leading to cell death can originate from more than one upstream initiator [35, 37]. Considering that the VZV RHIM appeared to have a limited effect in modulating TNF-directed necroptosis, we sought to determine whether it could influence cell death signalling pathways originating from other triggers in the context of infection. Recent research examining intrinsic immune responses to viral infection has demonstrated the nucleic acid sensing-protein ZBP1 as a key initiator of a variety of cell death pathways in response to MCMV, HSV-1, and IAV [27, 29, 43, 56]. To test if ZBP1 plays a role in initiating programmed cell death during VZV infection, we performed infectious centre assays using HT-29s that were engineered to express ZBP1 via lentivirus transduction, or empty vector transduced control HT-29s. Monolayers of ZBP1-expressing or control HT-29s were infected at a ratio of 1 infected HT-29 (wild type) to 10 uninfected (ZBP1/control) cells with the VZV BAC-derived parent strain or the RHIM mutated virus (VZV-RHIMmut) and plaque formation was assessed at 72 h post-infection. Cells were immunostained for VZV IE63 and infectious centres visualised by fluorescence microscopy.

The parental VZV virus (pOKA) formed plaques in both the ZBP1-expressing and control empty vector cells (Fig 4A), with no significant difference in plaque size seen between the two cell types (Fig 4B). VZV-RHIMmut readily formed plaques in the control empty vector HT-29s, which were on average the same size as plaques formed by pOKA (Fig 4B). However in contrast, VZV-RHIMmut virus spread was severely restricted in the ZBP1-expressing cells (Fig 4A), with plaque size significantly reduced (Fig 4B). The VZV-RHIMKO virus formed larger plaques on average in the empty vector control HT-29s, however similarly to the RHIM mutant virus, plaque size was also significantly reduced in the ZBP1-expressing cells (Fig 4). Given that during Influenza A infection ZBP1 has been shown to drive both apoptosis and necroptosis [29], and as our prior results suggested that VZV could not inhibit caspase 8, we tested if either or both pathways were being triggered during infection with the two mutant viruses. Monolayers of ZBP1-expressing or control empty vector HT-29s were infected at a ratio of 1:10 and then the pan-caspase inhibitor z-VAD-fmk (25 μM), or the MLKL inhibitor necrosulfonamide (NSA, 1 μM), was added to the cultures. After 72 h of infection, it appeared that the addition of necrosulfonamide had little impact on the spread of any of the viruses (Fig 4). Treatment of the ZBP1-expressing HT-29s with T+S+V confirmed that these cells retained the ability to undergo necroptosis (S3 Fig). In contrast, the addition of z-VAD-fmk led to an increase in average plaque size for all three viruses in ZBP1 expressing cells (Fig 4B), and increased the average plaque size for the two RHIM mutant viruses to similar levels as seen in the untreated empty vector control cells. This suggests that the ORF20 RHIM functions to inhibit ZBP1-induced apoptosis during VZV infection. This is the first time a viral RHIM protein has been associated with the inhibition of a non-necroptotic cell death pathway.

## RIPK3 is required for ZBP1-induced cell death to proceed in response to VZV infection

Previous reports describing ZBP1 induction of apoptosis in response to viral infection have indicated that, depending on cellular conditions, cell death can proceed in both a RIPK3-dependent and RIPK3-independent manner [29]. In order to elucidate the specific molecular pathways required for ZBP1-induced apoptosis in the context of VZV infection, we sought to determine whether RIPK3 was necessary for cell death. To this end, we performed infectious centre assays with ARPE-19 cells, a cell line known to be permissive to VZV infection that possesses a full complement of caspases but does not express RIPK3 [57]. For this assay, only the VZV parent strain and the VZV-RHIMmut were compared, since we had established that the IQIG core tetrad is crucial for the prevention of ZBP1-induced cell death (Fig 4). ZBP1-expressing ARPE-19 cells were engineered by lentiviral transduction and empty vector-transduced ARPE-19 cells were used as control. Monolayers of ZBP1-expressing or control ARPE-19 cells were infected at a ratio of 1 infected ARPE-19 (wild type) to 10 uninfected (ZBP1/control) cells, with the VZV BAC-derived parent strain or the RHIM mutated virus (VZV-RHIMmut), and plaque formation was assessed at 72 h post-infection. Cells were then immunostained for VZV IE63 and infectious centres were visualised by fluorescence microscopy (Fig 5). The parental VZV virus formed plaques in both the ZBP1 and control empty vector ARPE-19 cells to the same size and number. However, unlike in HT-29 cells expressing ZBP1, the VZV-RHIMmut was also able to form plaques equally as well in control and ZBP1-expressing ARPE-19 cells (Fig 5). The addition of z-VAD-fmk did not change the phenotype for either VZV parent or VZV-RHIMmut (Fig 5). These data indicate that the presence of ZBP1 and caspase 8 is insufficient for activation of cell death and RIPK3 is most likely necessary for cell death in response to VZV infection.

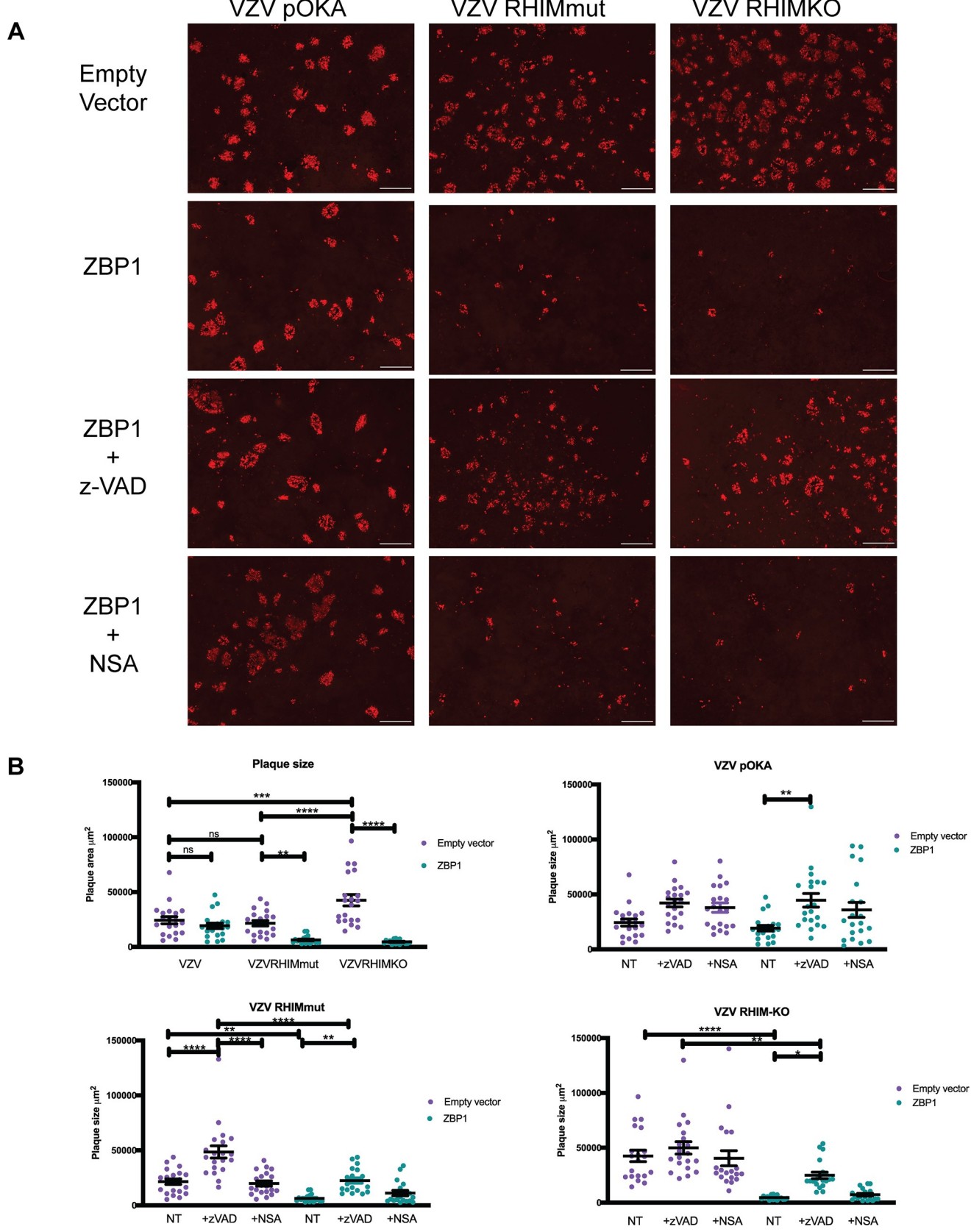

**Fig 4. ZBP1 restricts VZV-RHIMmut and VZV-RHIMKO spread in culture.** Empty vector or ZBP1 expressing HT-29s were inoculated with parental HT-29 cells either mock or infected with parent VZV (pOKA), VZV-RHIMmut or VZV-RHIMKO. z-VAD–fmk (25 μM) or necrosulfonamide (NSA; 1 μM) was added at the time of inoculation where indicated. (A) After 72 h cells were fixed and immunostained for VZV IE63 (red) to assess virus spread. Images are representative from at least 2 independent replicates. Scale bar indicates 500 μm. (B) The area of 18–20 plaques per virus (as indicated) was calculated using Zen 3.1 Blue edition (Zeiss), and statistical significance calculated using a one-way ANOVA. Line indicates mean and error bars represent standard error of the mean.

## VZV ORF20 interacts with RIPK3 and ZBP1 in cells, forming insoluble complexes with ZBP1

Herpesvirus inhibition of host RHIM protein signalling requires the direct interaction of host and viral proteins. For the viral inhibitors of necroptosis M45 and ICP6, both proteins interact directly with ZBP1 and RIPK3 through their respective RHIMs, an interaction that is ablated by a mutation of the core tetrad of the viral protein to AAAA [27, 38, 41]. We recently demonstrated that for M45, the interaction with RIPK3 or ZBP1 results in the generation of an insoluble amyloid structure containing both viral and host proteins, which acts to sequester the host

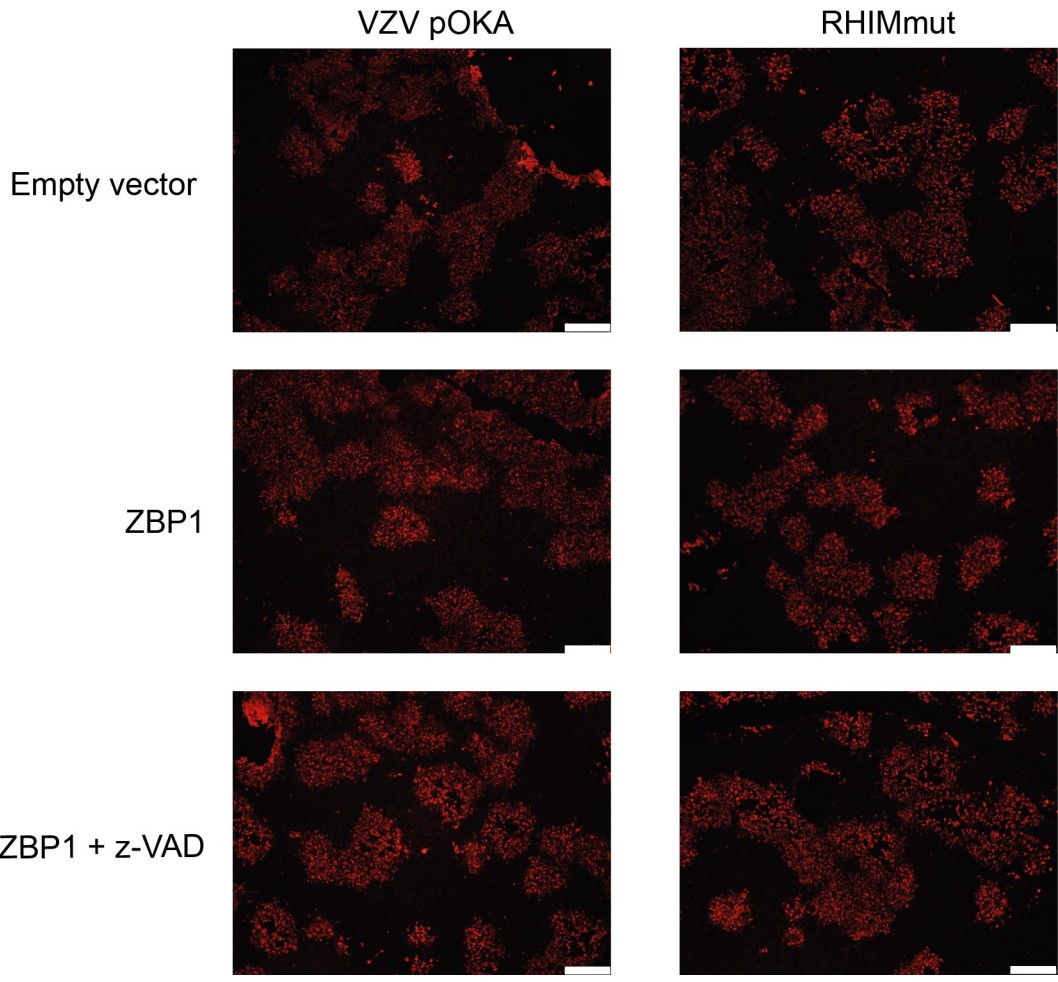

**Fig 5. ZBP1 does not restrict the spread of VZV or VZV-RHIMmut in RIPK3-deficient, apoptosis-capable cells.** Empty vector or ZBP1 expressing ARPE-19s were inoculated with parental ARPE-19 cells infected with parent VZV (pOKA) or VZV-RHIMmut. z-VAD-fmk (25 μM) was added at the time of inoculation where indicated. After 72 h cells were fixed and immunostained for VZV IE63 (red) to assess virus spread. Images are representative from at least 2 independent replicates. Scale bars represent 400 μm.

protein and prevent downstream functions [39]. We sought to determine whether ORF20 might interact with RIPK3 and ZBP1 in the same manner. 293T cells were co-transfected with plasmid constructs expressing either V5-tagged ORF20 or ORF20-RHIMmut and GFP or ZBP1-GFP or RIPK3-GFP, and interactions between the cellular and viral RHIM-containing proteins in both the soluble and insoluble fractions were investigated by immunoprecipitation (IP). Both ORF20 and ORF20-RHIMmut were found to interact with ZBP1 in the soluble (Fig 6A) and insoluble fractions (Fig 6B), indicating the formation of soluble complexes as well as insoluble supramolecular assemblies. The ability of the ORF20-RHIMmut to interact with ZBP1 was surprising but suggests that some residues within the RHIM, but outside of the core tetrad, can maintain an interaction between the proteins. Interaction between ORF20 and RIPK3, and ORF20-RHIMmut and RIPK3, was detected only in the soluble fraction derived from cell lysates (Fig 6C), despite multiple analyses of the insoluble fraction. We sought to examine interactions between ORF20 and RIPK1 by this method, and were unable to detect co-immunoprecipitation despite multiple attempts. We attempted to detect the interactions between ORF20 and both ZBP1 and RIPK3 during productive VZV infection of cells, but were unable to demonstrate these consistently, due to limitations of reagents available to perform these analyses.

## The ORF20 RHIM supports the formation of amyloid fibrils

The RHIMs within host and viral proteins have been shown to be responsible for functional amyloid fibril assembly by these proteins [21, 39, 58]. Having discovered that ORF20 and ZBP1 interact with each other to form large insoluble heteromeric complexes, we sought to determine whether these assemblies have the distinctive β-sheet rich substructure characteristic of an amyloid fibril [59]. The 114 N-terminal residues of ORF20, containing the RHIM, were expressed recombinantly in a fusion protein with His-tagged ubiquitin (Ub-ORF20$_{1-114}$). This construct spontaneously assembled into fibrils that exhibited the long, straight and unbranching morphology characteristic of amyloid fibrils (Fig 7A). A fusion construct containing the mCherry fluorophore (mCherry-ORF20$_{1-114}$), instead of ubiquitin, likewise exhibited typical amyloid morphology (Fig 7B). When incubated with the small-molecule amyloid sensors Thioflavin T (ThT) and Congo red, the presence of Ub-ORF20$_{1-114}$ fibrils gave rise to increased fluorescence emission at 485 nm with ThT and an increase in absorbance at 540 nm with Congo red, confirming the presence of a cross-β amyloid structure (Fig 7C and 7D). Protein assemblies with a fusion protein containing the AAAA mutation within the ORF20 RHIM (Ub-ORF20$_{1-114}$mut) displayed increased ThT emission at 485 nm and Congo red absorbance at 540 nm (Fig 7C and 7D), indicating that some cross-β substructure is present, however the rate of increase in ThT fluorescence was much slower than that for the WT RHIM construct. This construct also self-assembled into large amorphous aggregates that lacked a clear fibrillar structure (Fig 7E). Likewise, an mCherry-tagged ORF20 AAAA mutant protein (mCherry-ORF20$_{1-114}$mut) formed amorphous structures lacking a clear fibrillar morphology (Fig 7F). In summary, the ORF20 RHIM is able to form amyloid assemblies with characteristic morphology and structure and the AAAA mutation to the core tetrad impairs the formation of a regular, extended fibrillar structure.

## The core tetrad of the ORF20 RHIM controls the size and morphology of complexes formed with RIPK3 and with ZBP1

Following the observed interactions between ORF20 and human RIPK3 and ZBP1 in cells, we further investigated the nature of complexes formed between these proteins *in vitro*. Single molecule fluorescence confocal spectroscopy was used to confirm interactions at a single molecule level. This method uses lasers focused on a very small confocal volume (250x250x800 nm)

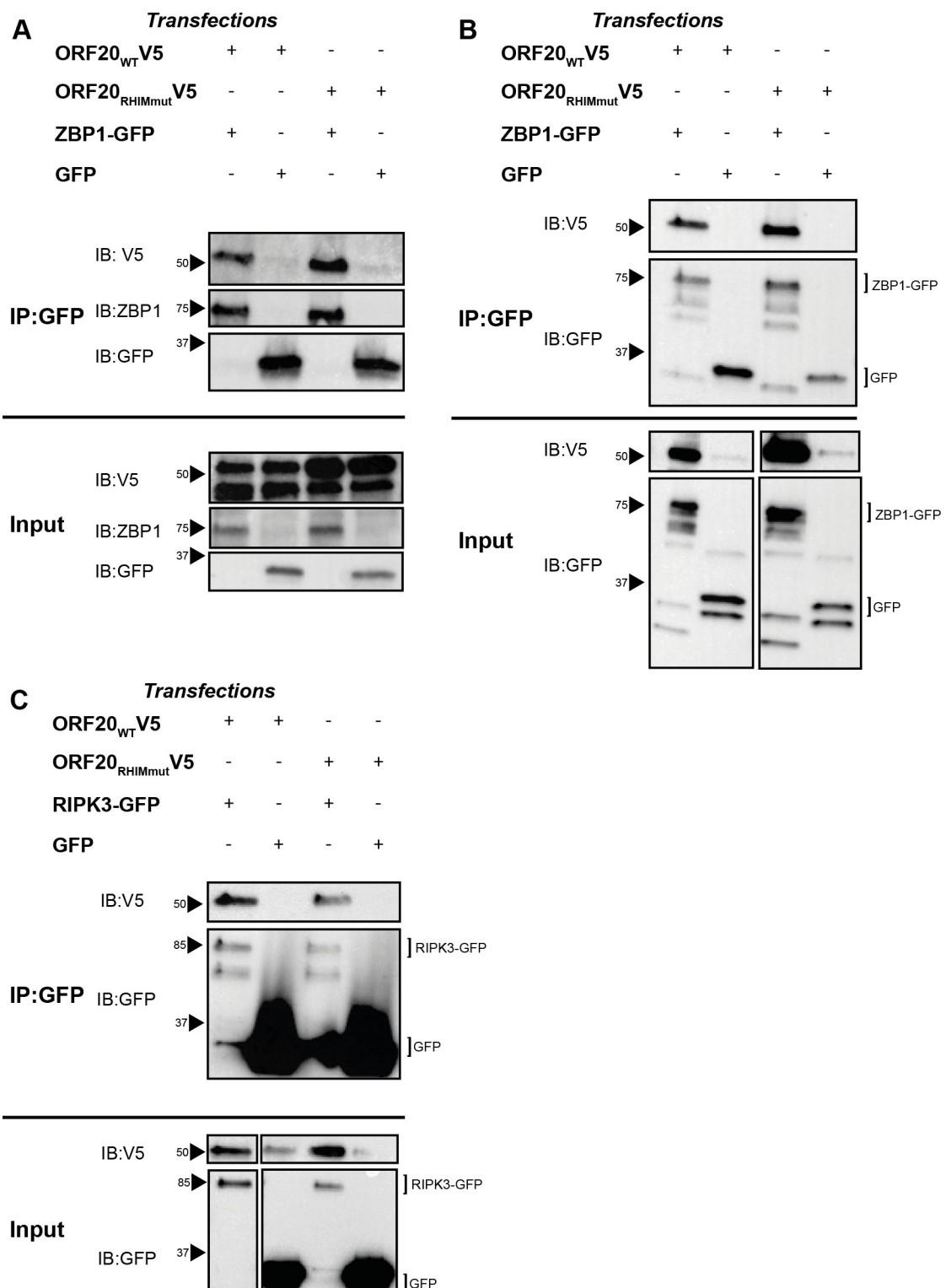

**Fig 6. ORF20 interacts with human RHIM proteins associated with cell death signalling.** 293T cells were transfected with ORF20 constructs and ZBP1-GFP or GFP alone. Cells were harvested and immunoprecipitation (IP) performed on the soluble and insoluble fractions using GFP as bait. Western blot (IB) was performed on the immunoprecipitated and input cell lysates. (A) Immunoprecipitation of ZBP1-GFP in the soluble fraction of cell lysates. (B) Immunoprecipitation of ZBP1-GFP in the insoluble fraction of cell lysates. (C) Immunoprecipitation of RIPK3-GFP in the soluble fraction of cell lysates. Each blot is representative of at least 2 independent biological replicates. Arrows indicate protein size markers.

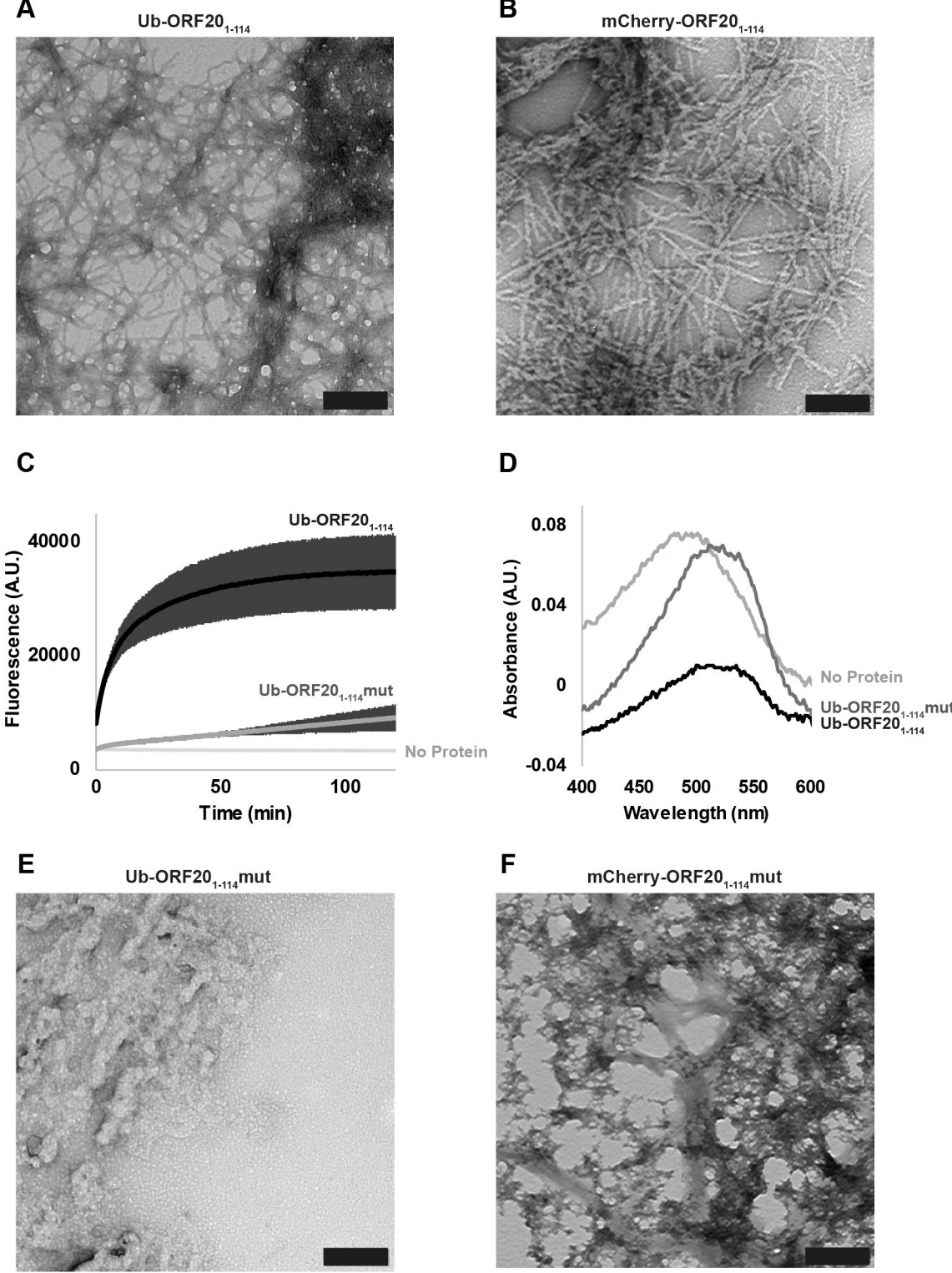

**Fig 7. The ORF20 RHIM forms homomeric amyloid structures.** (A) Transmission electron microscopy image of structures formed by Ub-ORF20$_{1-114}$. Scale bar represents 200 nm. (B) Transmission electron microscopy image of structures formed by mCherry-ORF20$_{1-114}$. Scale bar represents 200 nm. (C) ThT fluorescence over time of Ub-ORF20 constructs after dilution from 8 M urea into assembly buffer. Buffer sample contains equimolar ThT but no protein. Curves are derived from three independent replicates. Error bars indicate standard deviation. (D) Absorbance spectra of solutions containing Congo red and Ub-ORF20$_{1-114}$ and Ub-ORF20$_{1-114}$mut after dialysis against assembly buffer. Buffer refers to a Congo red sample in assembly buffer with no protein. (E) Transmission

electron microscopy image of structures formed by Ub-ORF20$_{1-114}$mut. Scale bar represents 200 nm. (F) Transmission electron microscopy image of structures formed by mCherry-ORF20$_{1-114}$mut. Scale bar represents 200 nm.

to detect fluorescent signals from two different proteins, labelled with two different fluorophores [60] (Fig 8A). Coincidence of fluorescent signals in mixtures of proteins at low concentration indicates the formation of hybrid complexes containing both protein partners. Fluorescently-labelled fusion proteins were prepared, with YPet or mCherry fused to host or viral proteins respectively. We observed large signals in the fluorescence intensity traces for

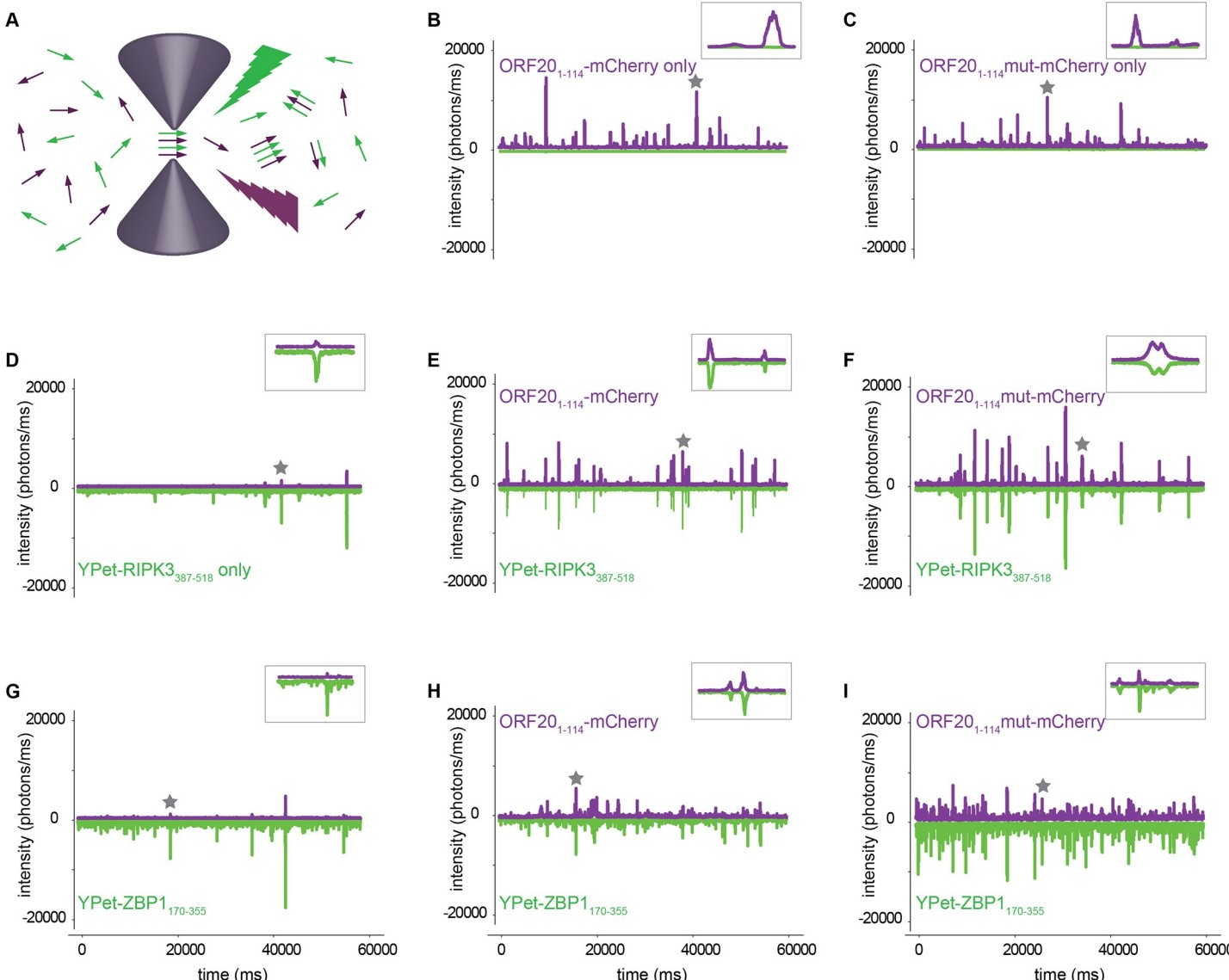

**Fig 8. The VZV ORF20 RHIM forms heteromeric complexes with RHIMs from RIPK3 and ZBP1.** (A) Schematic representation of fluorescence detection from dilute solutions containing two different fluorophores in a nanolitre confocal volume. Samples are excited simultaneously with two overlapping lasers and emission from the YPet and mCherry fluorophores is recorded separately. Coincident bursts in the two channels indicate formation of heteromeric complexes containing two different proteins (B-I). Representative fluorescent time traces collected from ORF20$_{1-114}$-mCherry, ORF20$_{1-114}$mut-mCherry, YPet-RIPK3$_{387-518}$ or YPet-ZBP1$_{170-355}$ fusion proteins, alone or mixed in pairs under conditions that allow co-assembly. The proteins present in each mixture are indicated for each part of the figure. Inserts show detail of 1 s dual fluorescence recording, indicated by star on the full time trace.

ORF20$_{1-114}$, ORF20$_{1-114}$mut, RIPK3$_{387-518}$ and ZBP1$_{170-355}$ proteins when analysed alone, indicating homo-oligomer formation by these proteins (Fig 8B, 8C, 8D and 8G). When the ORF20 proteins were mixed with the human proteins under assembly competent conditions both ORF20$_{1-114}$ and ORF20$_{1-114}$mut were observed to form hetero-oligomers with RIPK3$_{387-518}$ and ZBP1$_{170-355}$, indicated by coincidence of fluorescent signals in both channels (Fig 8E, 8F, 8H and 8I). We also performed confocal spectroscopy experiments to analyse interactions between ORF20 and RIPK1 (S4A Fig). In accordance with our inability to detect an interaction between ORF20 and RIPK1 by co-immunoprecipitation, no interaction between ORF20-$_{1-114}$-mCherry and the minimum region of RIPK1 that has been shown to be required to interact with RIPK3 (YPet-RIPK1$_{497-583}$) [21] was detected. Likewise, no interaction between YPet-RIPK1$_{497-583}$ and ORF20$_{1-114}$mut-mCherry was detected (S4A Fig). These data indicate that ORF20 and RIPK1 are unlikely to interact directly.

Considering that we had previously established that the interactions between ZBP1 and ORF20 were the most relevant in a biological setting, we analysed the sizes of the heterocomplexes formed between these constructs. Photon counting histograms reflect the distribution of particle sizes detected in the confocal volume. Analysing the signal for mCherry from the single molecule traces, we determined the size distribution of ORF20$_{1-114}$ wild type and ORF20$_{1-114}$mut oligomers alone and in combination with ZBP1$_{170-355}$. For ORF20 RHIM wild type, there was no difference in the size of particles formed by ORF20$_{1-114}$ with or without ZBP1$_{170-355}$ (S5A Fig). This indicates that the ORF20 RHIM is capable of forming large molecular assemblies by itself and in concert with ZBP1. ORF20$_{1-114}$mut-mCherry formed smaller structures than its wild type counterpart (S5B Fig). Further, together ORF20$_{1-114}$mut-mCherry and ZBP1$_{170-355}$ form structures smaller than the mutant ORF20 alone, and substantially smaller than the combination of wild type ORF20 and ZBP1, reflected in the histograms and in fluorescence correlation analysis of the single molecule traces (S5C Fig).

We also assessed interactions between ORF20 and mutant forms of YPet-ZBP1, where either the first RHIM of ZBP1 was mutated to AAAA (YPet-ZBP1$_{170-355}$mutA) or the second RHIM was mutated to AAAA (YPet-ZBP1$_{170-355}$mutB) (S4B Fig). Mutation of the first RHIM of ZBP1 reduces the ability of the protein to self-assemble into homomeric amyloid structures and little co-assembly of ORF20 with YPet-ZBP1$_{170-355}$mutA was observed. Mutation of the second RHIM does not affect self-assembly and YPet-ZBP1$_{170-355}$mutB interacted strongly with ORF20 (S4B Fig). These findings are in agreement with the reported effects of RHIM mutations on ZBP1 activity in cells, where the first RHIM is primarily associated with interaction and assembly of RHIM-based structures [27, 41, 43].

## Mixtures of ORF20 and ZBP1 form heteromeric amyloid assemblies with fibrillar morphology

We recently proposed that M45 successfully inhibits host RHIM:RHIM interactions by trapping host proteins in alternative stable hybrid amyloid structures [39]. In order to determine if a similar mechanism was occurring during ORF20-ZBP1 interactions, we examined mixtures of Ub-ORF20$_{1-114}$ and Ub-ORF20$_{1-114}$mut with YPet-ZBP1$_{170-355}$ via electron microscopy (Fig 9A). YPet-ZBP1$_{170-355}$ alone forms short, clustered fibrils <200 nm in length, confirming the amyloidogenic nature of ZBP1 previously reported [21]. In combination with wild type ORF20, however, large and dense networks of fibrils were evident (Fig 9A), similar to those formed between M45 and RIPK3 [39]. When ZBP1 was incubated with ORF20$_{1-114}$mut, only sparse amorphous aggregates were observed (Fig 9A). The fibrils formed by wild type ORF20 and ZBP1 proteins were further examined by confocal microscopy, using ORF20$_{1-114}$-mCherry and YPet-ZBP1$_{170-355}$ fluorescent constructs (Fig 9B). This demonstrated that the

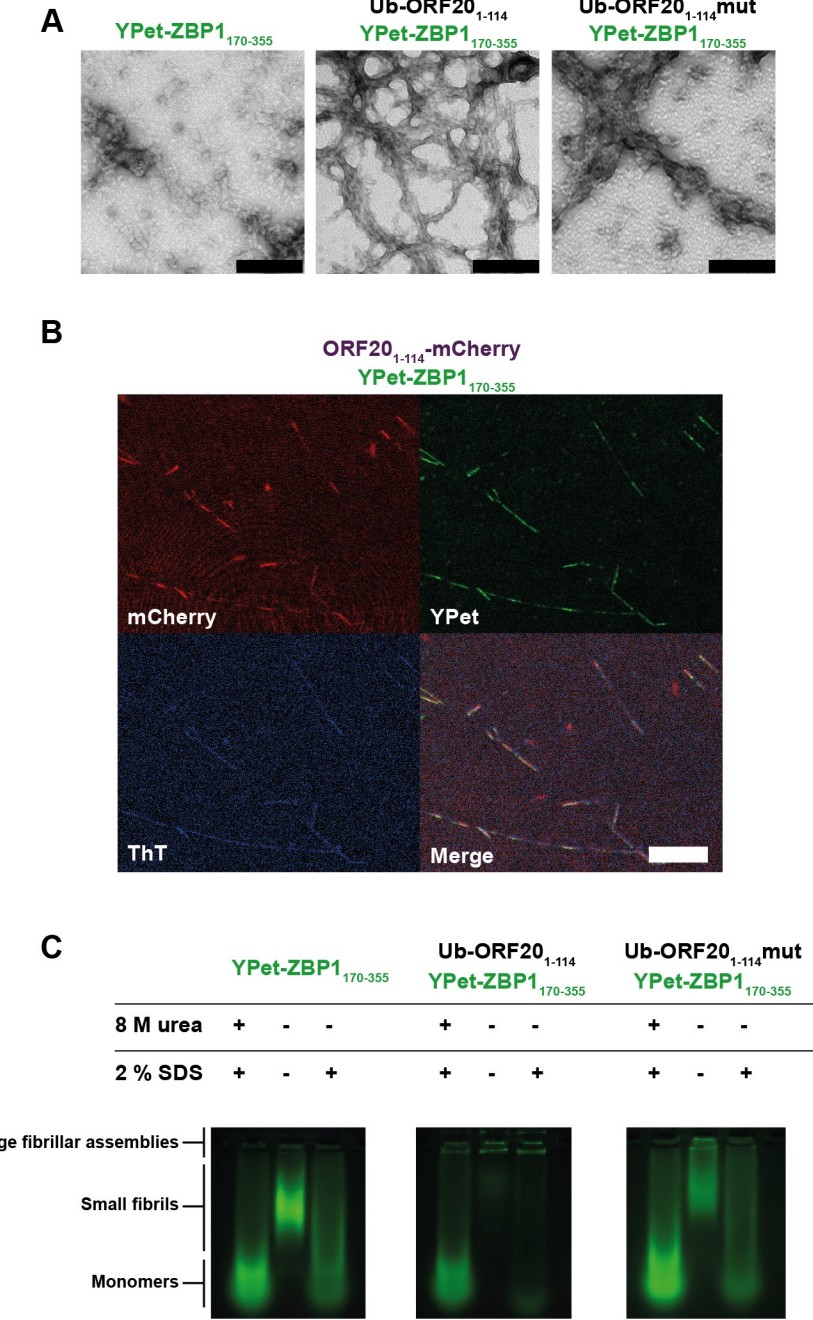

**Fig 9. The VZV ORF20 RHIM forms heteromeric amyloid fibrils with ZBP1, which are dependent on the core tetrad for stable higher-order assembly.** (A) Transmission electron micrograph of protein assemblies formed from combinations of Ub-ORF20$_{1-114}$, Ub-ORF20$_{1-114}$mut, and YPet-ZBP1$_{170-355}$ fusion proteins. Scale bar represents 200 nm. (B) Confocal microscope images of heterofibrils formed from a mixture of ORF20$_{1-114}$-mCherry and YPet-ZBP1$_{170-355}$ in ThT-containing assembly buffer. Scale bar represents 20 μm. (C) The IQIG core tetrad from ORF20 is required to trap ZBP1 in insoluble, detergent-stable aggregates. SDS-AGE analysis of monomeric or assembled forms of Ub-ORF20$_{1-114}$, Ub-ORF20$_{1-114}$mut, and YPetZBP1$_{170-355}$, or combinations thereof. Monomeric proteins were maintained in 8 M urea before electrophoresis. Assembled samples were incubated with either water or 2% SDS at room temperature for 10 min before electrophoresis. Identity and treatment of protein indicated above each sample.

proteins were co-incorporated within fibrils. The amyloid nature of these heteromeric fibrils was confirmed by observation of ThT signal along the length of these fibrils (Fig 9B). Overall, these results suggest that the ORF20 RHIM is able to recruit the human ZBP1 RHIM into fibrillar assemblies, and that branching networks are dependent on the core tetrad of the RHIM.

## The core tetrad of the ORF20 RHIM is important for the stability of the ORF20-ZBP1 fibrillar assemblies

Differences in the size and stability of assemblies formed between ORF20 and ZBP1 were examined by sodium dodecyl sulfate agarose gel electrophoresis (SDS AGE). Assemblies were prepared from each protein alone or in combination and then incubated with or without 2% SDS before electrophoresis alongside monomeric forms of the proteins (Fig 9C). YPet-ZBP1 RHIM was observed to form oligomers with retarded mobility, relative to monomer (Fig 9C), corresponding to the <200 nm-long fibrils observed by TEM (Fig 9A). Co-assembly with non-fluorescent Ub-ORF20$_{1-114}$ results in a retention of the YPet-ZBP1$_{170-355}$ in the wells, indicating that ZBP1 is sequestered by the VZV RHIM into large assemblies that are too large to enter the gel (Fig 9C). Unlike the fibrils formed by ZBP1 alone, which are depolymerised upon incubation with 2% SDS, these very large heteromeric amyloid assemblies are stable and resistant to depolymerisation, as demonstrated by the unchanged mobility of the ZBP1-ORF20-containing fibrils following detergent treatment (Fig 9C).

Co-assembly of the ORF20$_{1-114}$mut with ZBP1 did not generate stable oligomers (Fig 9C). Although oligomeric structures are formed when ZBP1 and Ub-ORF20$_{1-114}$mut interact (Fig 9A), corresponding to the amorphous aggregates observed by electron microscopy, these are not as large as those that contain ZBP1 and Ub-ORF20$_{114}$-WT (Fig 9C). Additionally, these assemblies are not resistant to treatment with 2% SDS (Fig 9C). These results demonstrate that the heteromeric complexes formed between ZBP1 and the ORF20-RHIMmut are less stable than those formed between ZBP1 and WT ORF20, confirming the important role of the IQIG sequence in the functional interactions between the RHIMs of ZBP1 and ORF20.

## Discussion

This study identifies a new viral RHIM encoded by VZV ORF20. Whereas the viral RHIMs previously identified in other herpesviruses (MCMV, HSV-1 and -2) have been shown to be important for the suppression of necroptosis [27, 41], the VZV ORF20 RHIM is unique in that its primary role appears to be the inhibition of ZBP1-induced apoptosis. Further, we show the amyloid nature of the VZV ORF20 RHIM, and given the amyloidogenic nature of the ZBP1 RHIM demonstrated here and elsewhere [21], suggest that ORF20 functions as a decoy RHIM, by sequestering ZBP1 within alternative amyloid assemblies and impairing its ability to interact with RIPK3. We propose that the RHIM within VZV ORF20 provides a novel mechanism for the prevention of ZBP1-induced extrinsic apoptosis during VZV infection (Fig 10). Our results indicate that ZBP1 senses VZV infection within the cell. During a productive infection ORF20 proteins, either incoming from infective capsids and/or newly-expressed, interact with ZBP1, and through RHIM-dependent mechanisms, form stable heteromeric amyloid assemblies which supresses the usual signalling capabilities of the host protein. These results advance our understanding of viral modulation of RHIM signalling and demonstrate diversity with respect to the inhibitory roles of viral RHIMs.

The viral RHIM identified in VZV ORF20 appears to be a conserved motif within the *Varicellovirus* genus, with other members encoding at a minimum the core I/V-Q-I/V-G sequence within the N-terminal of the capsid triplex subunit 1. Whether these other viruses possess the

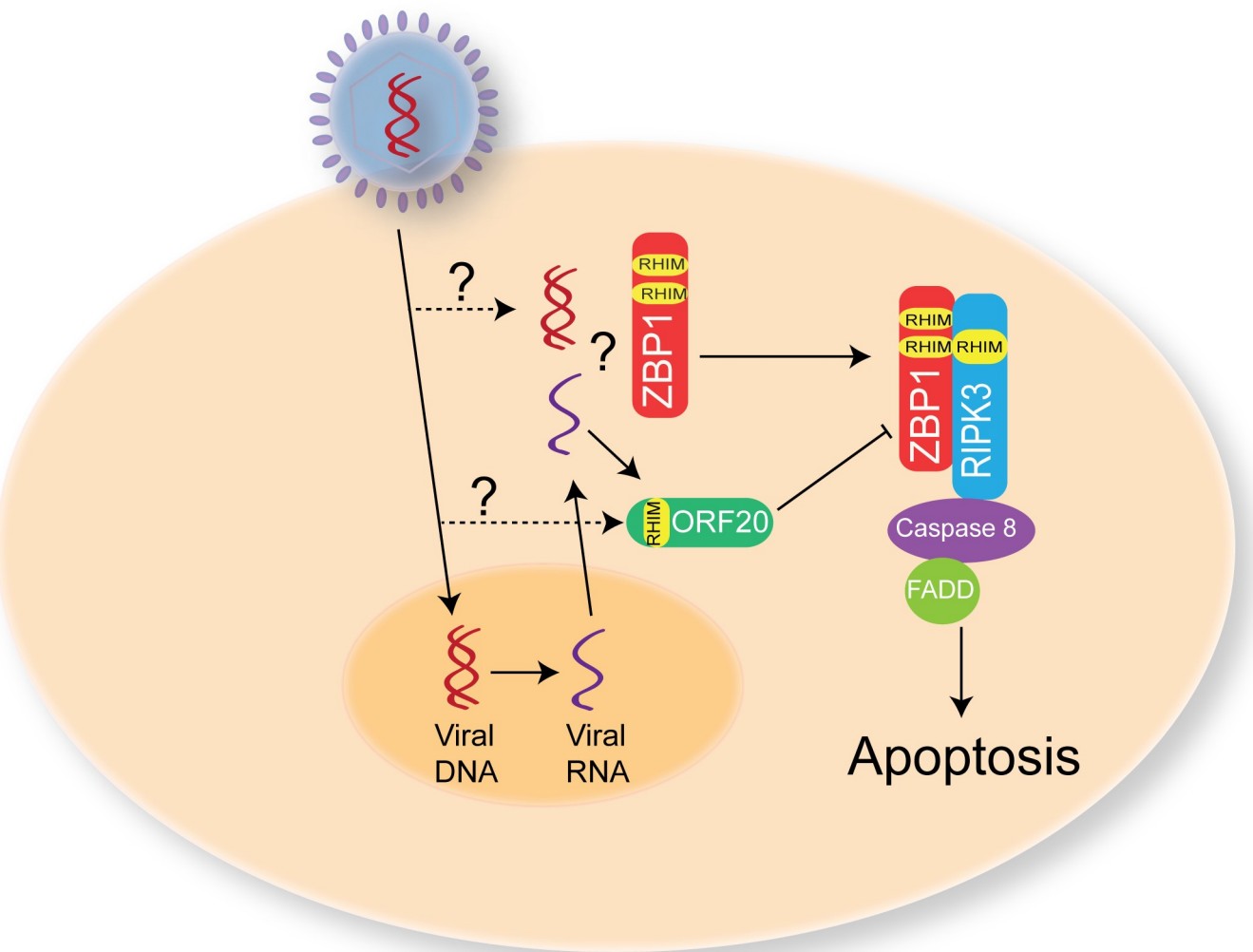

**Fig 10. Proposed mechanism for VZV ORF20 RHIM during VZV infection.** Upon VZV infection of a host cell the viral DNA is transported to the nucleus with some potentially leaked into the cytoplasm. Viral DNA is then transcribed into RNA and exported from the nucleus. ZBP1 can sense either VZV RNA or DNA and then via RHIM:RHIM interactions activate RIPK3 to drive apoptosis. However, the ORF20 RHIM, either from incoming viral capsids, or following *de novo* synthesis can interact with ZBP1 to prevent it driving apoptosis and thus prolong the survival of the infected cell to allow for the complete viral replication cycle to occur.

ability to modulate RHIM signalling in their natural hosts via this motif remains to be elucidated. VZV is an exclusively human pathogen so it would also be of interest to assess if the ORF20 RHIM has inhibitory functions in non-human cell types, or if it may drive necrosome assembly in a similar manner to HSV-1 ICP6 and HSV-2 ICP10 in murine cells [40, 42]. These opposing functions of the HSV RHIMs lead to the suggestion that necroptosis can act as a species-barrier to infection [41], and VZV infection is known to be inhibited post-entry from replicating in most cells of non-human origin [61]. In addition, cell death is also important in mediating the species-specific nature of CMV infection [62], however we could not find any evidence of a RHIM-like motif in the capsid triplex 1 protein of any human beta or gamma herpesviruses.

ICP6 and ICP10 also bind and inhibit caspase 8 as a means of blocking extrinsic apoptosis driven by death receptor signalling, which then opens the so-called necroptotic "trap door"

[35]. Thus, ICP6 and ICP10 can simultaneously inhibit both TNF-driven apoptosis and necroptosis. We have shown that VZV-infected HT-29s were protected from TNF-driven necroptosis as evidenced by an increase in cell survival and a lack of pMLKL in infected cells following treatment. This inhibition of necroptosis could not be fully attributed to the ORF20 RHIM, implying that VZV encodes alternative mechanisms to inhibit TNF-induced necropto-sis. HCMV employs a non-RHIM based mechanism to inhibit TNF-induced necroptosis [47] and VZV could utilise a similar mechanism. Curiously, however, VZV infection did not appear to protect against TNF-driven apoptosis. This raises the question as to why VZV encodes a necroptosis inhibitor. It has been shown that infection of murine macrophages with MCMV lacking both M36 and M45, and thus unable to inhibit caspase 8 and RHIM signalling, leads to a highly inflammatory cell death program involving apoptosis followed by secondary necroptosis [63]. This demonstrates that the inhibition of caspase 8 is not always a prerequisite for necroptosis to occur and therefore it is possible that in certain cell types VZV infection directly triggers necroptosis, which the virus has subsequently evolved to inhibit. Previously we have shown that VZV ORF61 can inhibit TNF-driven NF-κB activation via its E3 ubiquitin ligase domain [64]. Thus VZV can successfully inhibit two of the three possible outcomes of TNF signalling. In addition to the requirement for an intact RHIM, a conserved motif in the C-terminal region of M45 and ICP6, which bind RIPK1, has been shown to be necessary for the inhibition of TNF-induced necroptosis [65]. Our finding, that unlike HSV and MCMV, the RHIM in VZV is not associated with a ribonucleotide reductase domain, may explain the difference in function of the VZV RHIM versus that of M45 and ICP6. Currently it is not known if the C-terminal region of M45 and ICP6 are involved in the ZBP1 necroptotic pathway.

Our data indicates that the RHIM-based mechanism that VZV utilises to inhibit apoptosis specifically targets ZBP1. Although we found that ORF20 and RIPK3 were capable of interact-ing in a number of contexts (Fig 6, Fig 8), our studies of cellular protein interaction indicated that ORF20 is capable of sequestering ZBP1 into large, insoluble supercomplexes (Fig 6), whereas ORF20 and RIPK3 were only observed to occur in soluble contexts in cells, which indicates either dimeric or small oligomeric interaction structures formed by the two proteins. The reason for this difference may be due to a low concentration of protein in the RIPK3 experiments, as transfection with the RIPK3-GFP construct was observed to impact cell viabil-ity and the protein could only be detected at low levels. Alternatively, this may indicate that ORF20 only forms large insoluble complexes with ZBP1. This may explain why ORF20 is inca-pable of inhibiting RIPK3-directed necroptosis pathways, instead targeting ZBP1 as an initia-tor specifically.

Studies have shown that ZBP1 can simultaneously drive apoptosis and necroptosis follow-ing sensing of IAV [29]. In addition, ZBP1 can also drive pyroptosis in IAV infected murine macrophages [56]. We found that in HT-29s expressing ZBP1, VZV replication was restricted when the ORF20 RHIM was mutated or removed. The biophysical analyses reported here indi-cate that the mutation results in complexes of a reduced size and stability. We propose that these complexes containing the RHIM mutant are unable to effectively and stably sequester and inhibit all of the ZBP1 expressed in our system during VZV-RHIMmut infection. The addition of the pan-caspase inhibitor z-VAD-fmk rescued viral plaque formation by both mutant viruses, whereas the addition of necrosulfonamide had no impact, implying that ZBP1 was initiating apoptosis following infection with VZV and that the ORF20 RHIM inhibits this during wild type infection, presumably by interacting directly with ZBP1 (Fig 10). Given that enones such as necrosulfonamide are highly reactive, the half-life of necrosulfonamide in cul-ture is likely to be limited and thus we cannot rule out that some necroptosis also occurred. It is interesting that the addition of z-VAD-fmk rescued viral replication rather than triggering

necroptosis, and this further implies that VZV may encode another specific inhibitor of necroptosis.

Like the host RHIM-containing proteins RIPK1, RIPK3 and ZBP1, ORF20 is able to self-assemble into homo-oligomeric protein fibrils with an amyloid structure and it forms amyloid-based hetero-oligomeric fibrils. The co-assembly of ORF20 and ZBP1 resulted in the formation of very long heterofibrils, strikingly different to fibrils formed by either protein alone, and containing both proteins along the fibril length. Confocal microscopy and SDS AGE experiments demonstrated that the tetra-alanine RHIM mutant of ORF20 retained the ability to interact with the RIPK3 and ZBP1 RHIMs *in vitro*. Similarly, by co-immunoprecipitation we found that wild type and RHIM mutant ORF20 could interact with ZBP1. This contrasts with the other viral RHIMs identified, which lose the ability to interact with cellular RHIMs when the core tetrad is mutated to AAAA [38, 41]. However, we have previously demonstrated that some ability to form amyloid assemblies is also retained when the core tetrad of RIPK3 is mutated to AAAA [39]. Hence we conclude that in the ORF20-RHIMmut, as in RIPK3-R-HIMmut, residues outside the core tetrad of IQIG can form inter-protein contacts with other RHIMs to generate complexes with elements of cross-β structure. However, we also observe that these mutant ORF20-RHIMmut-ZBP1 structures lack the distinct structure of the ORF20WT-ZBP1 amyloid fibrils and are smaller and less stable than the ORF20WT-ZBP1 complexes, and this may explain why the VZV-RHIMmut virus was unable to inhibit ZBP1-induced cell death in HT-29s. The atomic resolution structure of the host RIPK1-RIPK3 RHIM amyloid necrosome fibril core demonstrates that highly specific and selective residue-residue interactions define the stoichiometry and morphology of productive host necrosome structures [58]. Therefore, we propose that the wild type complex formed by ORF20 and ZBP1 has a particular stable fibrillar architecture that sequesters ZBP1 in a manner that prevents the latter from signalling, while the interaction between the ORF20 mutant and ZBP1 lacks the stability and order necessary to inhibit ZBP1 function. Further investigation will be required to determine the full extent of the interacting residues within this motif, which may differ for different pairs of RHIM-containing proteins. The identification of new RHIM sequences such as that from ORF20 will aid in determining the minimum sequence requirements for functional activating and/or inhibitory RHIMs and define the role of the core tetrad in the assembly of the functional amyloid necrosome.

Studies in mice have shown that ZBP1 is expressed in keratinocytes and can drive necroptosis in the absence of RIPK1 expression [66, 67]. As keratinocytes are a primary target of VZV during varicella and herpes zoster [68–70], it will be relevant to determine whether the ORF20 RHIM is essential for virus replication in these skin cells. We have previously reported that VZV ORF63 can inhibit apoptosis in neurons [71, 72]. It has not yet been determined whether ZBP1 is expressed in sensory neurons, but it would also be of interest to assess if inhibition of ZBP1 by VZV ORF20 plays a role in neuronal infection and the establishment of latency.

ORF20 is part of the capsid triplex complex in addition to ORF41 [73]. Currently we do not know if *de novo* ORF20 synthesis is required for ORF20 to interact with ZBP1, or if the capsid-associated protein on incoming viral particles can perform this role (Fig 10). Herpesvirus capsid assembly occurs in the nucleus of infected cells, however transient transfection of ORF20 in cells results in a largely cytoplasmic distribution of the protein and it is thought that binding to other capsid proteins is required to facilitate nuclear entry [73]. Therefore, ORF20 could interact with ZBP1 in the cytoplasm of infected cells. Alternatively, a recent report has suggested that activation of RIPK3 by ZBP1 can occur in the nucleus [74] and therefore ORF20 could interfere with this process during capsid assembly. It is also unclear whether, ZBP1 senses VZV RNA, DNA, or a combination of the two, although recent reports indicate that ZBP1 is most likely a sensor of RNA [29, 30].

Together our results show that VZV encodes a novel viral RHIM with a primary function of inhibiting ZBP1-induced apoptosis. Our results suggest that VZV ORF20 forms an amyloid assembly with ZBP1 via RHIM:RHIM interactions to inhibit cell death during VZV infection. This is also the first report of VZV-inhibition of TNF-induced necroptosis. These results increase our understanding of viral manipulation of host RHIM signalling pathways, which extends further than just necroptotic signalling outcomes.

## Materials and methods

### Cells and virus

HT-29 adenocarcinoma cells (a kind gift from A/Prof. Susan McLennan), human foreskin fibroblasts (HFF-1), ARPE-19 and Human embryonic kidney (HEK) 293T cells (ATCC) were maintained in Dulbecco's Modified Eagle's Medium (Lonza) supplemented with 10% fetal calf serum (Serana) and pencillin/streptomycin (ThermoFisher Scientific). VZV rOKA was a kind gift from Prof Ann Arvin (Stanford University, CA, USA) and was propagated in a cell-associated manner in ARPEs and then HT-29s. When infected cell monolayers displayed a cytopathic effect in 50–75% of cells, cells were trypsinised (trypsin-EDTA, Gibco), and added to uninfected monolayers of cells at a ratio of between 1:6–1:10, as described previously [49].

### Reagents and antibodies

The following antibodies were used for immunofluorescence: VZV IE62 and gE:gI glycoprotein complex (clone SG1, Meridian Lifescience), VZV pORF29 and IE63 (kindly provided by Prof Paul Kinchington, University of Pittsburg, PA, USA) and phosphorylated MLKL (Abcam) or mouse IgG1, normal rabbit serum or RabMab IgG as isotype controls. Bound primary antibodies were detected with Alexa Fluor 488 or 594 conjugated donkey anti-mouse or anti-rabbit secondary antibodies (Life Technologies).

Antibodies used for immunoblotting: rabbit anti-phosphorylated and total MLKL (Abcam), mouse anti-VZV ORF20 (Capri antibodies), rabbit anti-actin, mouse anti-V5 and rabbit anti-ZBP1 (Sigma-Aldrich), with secondary HRP conjugated donkey anti-mouse or donkey anti-rabbit antibodies (Santa Cruz).

### Flow cytometry for quantitation of VZV infected cells

Mock and VZV infected HT-29 cells were trypsinised and washed 2 x in FACS buffer (PBS containing 1% fetal calf serum and 10 mM EDTA). Cells were then stained for the VZV gE:gI glycoprotein complex (clone SG1, Meridian Lifescience [conjugated in house to dylight 488]) or an isotype control, for 20 mins at 4 °C. Cells were then washed 2 x in FACS buffer and then incubated in Cytofix fixation buffer (BD Biosciences). Cells again washed in FACS buffer prior to performing flow cytometry using a BD Fortessa flow cytometer and data analysis was performed using FlowJo software (Tree Star).

### Plasmid constructs

ORF20 or ORF20-RHIMmut were PCR amplified (Phusion, NEB), a V5 tag incorporated and inserted into the pCDH-EF1 vector, and verified via restriction enzyme digests and sequencing (Garvan Institute, Sydney Australia). GFP tagged ZBP1 was expressed in pEGFP-N1, provided by Prof Elizabeth Hartland (Hudson Institute of Medical Research, Australia), and pEGFP-C1 (Clontech) was used as a GFP only control.

## Cell viability assay

Uninfected and VZV-infected HT-29s were seeded in 96 well plates ($10^4$ cells/well) and allowed to adhere for 7 h before treating with combinations of TNF (R and D), BV-6 (Selleck-chem), z-VAD-fmk (AdooQ Bioscience and R and D), and necrostatin-1 (Selleck Chemicals) or DMSO only (Sigma-Aldrich) as control for 17–18 h. Cell viability was assessed by measuring levels of intracellular ATP, using the CellTitre-Glo2 Assay (Promega). Data was expressed as percentage of cell survival relative to the DMSO only control. Luminescence was measured using an Infinite M1000 Pro plate reader (TECAN). A minimum of 3 independent replicates was performed for each experiment.

## Immunofluorescence staining

For assessment of the phosphorylation of MLKL and VZV antigen expression, uninfected and VZV-infected HT-29s were seeded onto coverslips and allowed to adhere overnight. Cells were then treated with T+S+V, or DMSO as a control, for 7–8 h. Cells were then washed in PBS and fixed with cytofix (BD). Cells were permeabilised with ice cold methanol for 10 mins, then washed in Tris buffered saline (TBS). Blocking was performed using 20% normal donkey serum (NDS, Sigma-Aldrich) then primary antibodies (diluted in 10% NDS) incubated overnight at 4 °C.

For detection of VZV antigens alone cells were fixed with Cytofix (BD biosciences), washed in PBS and permeabilised with 0.1% Triton X-100 for 10 mins (Sigma-Aldrich). Blocking was performed using 20% normal donkey serum (NDS, Sigma-Aldrich) then primary anti-VZV antibodies (diluted in 10% NDS) incubated for 1 hr at room temperature. In both cases bound primary antibodies were detected using species-specific Alexa Fluor 488 or 594 conjugated secondary antibodies. Cells were then mounted in Prolong Gold antifade containing 4',6-Diamidino-2-Phenylindole, Dihydrochloride (DAPI) (Life Technologies). Imaging was performed using a Zeiss Axioplan 2 upright microscope with an Axiocam camera (Carl Zeiss). Counting for pMLKL was performed by randomly imaging 10–20 non-overlapping regions of each slide and manually counting cells.

## Infectious centre assays

ZBP1-expressing or empty vector-transduced HT-29 or ARPE-19 cells were seeded onto coverslips and allowed to adhere overnight. Each cell type was then infected with either VZV- or VZV-RHIMmut by the addition of infected HT-29/ARPE-19 cells by cell-associated infection. z-VAD-fmk (R and D) or necrosulfonamide (Selleck Chemicals) was added to certain wells in order to probe rescue from cell death. Infected cells were incubated for 72 hours. Cells were then washed in PBS and fixed and immunostained for VZV IE63 as described above. To quantitate plaque size, the area of 18–20 plaques per virus was calculated using Zen 3.1 Blue edition (Zeiss).

## Generation of mutant VZV strains

Mutations to the VZV genome were performed utilising a bacterial artificial chromosome (BAC) containing the VZV pOKA genome [75]. Homologous recombination was performed using SW102 *E. coli* (kindly provided by Dr Russel Diefenbach) as previously described [76]. Following insertion of the selection cassette into the ORF20 gene, it was then replaced using an oligonucleotide containing the desired tetra-alanine substitution in the ORF20 RHIM core (IQIG to AAAA) or with an oligonucleotide with 20 amino acids encompassing the ORF20 RHIM removed (amino acids 21–40 inclusive). Following recombineering, BAC genomes

were subject to restriction enzyme digestion, and regions encompassing the mutations were sequenced (Garvan Institute, Sydney, Australia) to confirm successful incorporation of the mutation. BACs were purified using Nucleobond Xtra BAC (Machnery-Nagel), then transfected into ARPE-19 cells using Fugene HD (Promega) to recover infectious virus, which was then propagated as described above.

## Immunoprecipitations

293T cells were transfected using Fugene HD (Promega) as per the manufacturer's instructions. Cells were harvested in DISC lysis buffer (1% Triton X-100, 10% glycerol, 150 mM NaCl, 2 mM EDTA and 30 mM Tris, pH 7.5) containing protease inhibitor cocktail at 1:100 (Sigma Aldrich). Cell lysates were centrifuged at 16,000 x $g$ at 4˚C, and the supernatant retained as the 'soluble' fraction. The cell pellet was resuspended in a denaturing buffer (8 M urea, 20 mM Tris, pH 8.0), and incubated at 4˚C overnight by gentle agitation. This urea resuspension was retained as the 'insoluble' fraction. GFP-TrapA (Chromotek) was used to immunoprecipitate GFP-tagged proteins from these fractions as required, as per manufacturer's instructions. Bound and input fractions were resolved by Western blotting.

## Western blotting

For analysis of pMLKL, cells collected and boiled in Laemmli sample buffer (BioRad) containing 1% 2-mercaptoethanol (Sigma Aldrich). Protein lysates were electrophoresed using 10% Mini-PROTEAN TGX gels (BIO-RAD) and transferred to PDVF membranes (Millipore) using the Trans-blot turbo system (BIO-RAD). Following transfer, membranes were washed in PBS then blocked in PBS containing 3% bovine serum albumin (Sigma Aldrich). Primary antibodies were applied overnight and bound primary antibodies were detected by probing with species specific HRP conjugate secondary antibodies in 5% skim milk. Detection was then performed via chemiluminescence (GE healthcare life sciences) and a Chemidoc XRS + system (BIO-RAD).

## Generation of ZBP1-expressing HT-29s

ZBP1 was cloned from the ZBP1-GFP construct, and inserted into pCDH-MCS-EF1-neo lentivector. Correct insertion was validated by sequencing (Garvan Institute, Sydney, Australia). Lentivirus for ZBP1 or empty vector control were made in 293T cells by co-transfecting the lentivectors with psPAX2 and pMD2G. HT-29s were then transduced with the lentivirus particles, and after 3 days, G418 was added for the subsequent 10 days to select for successfully transduced cells. Expression of ZBP1 was confirmed by Western blotting as described above.

## Expression of RHIM fusion proteins

Wild type or AAAA versions of the RHIM-containing regions of human RIPK3 (Q9Y572; residues 387–518), human ZBP1 (Q9H171; residues 170–355) and VZV (strain Dumas) ORF20 (P09276; residues 1–114) were expressed as His-tagged N- or C-terminal YPet, mCherry or ubiquitin fusion proteins (S6 Fig) in *E. coli* BL21(DE3) pLysS (Novagen) grown in LB media containing ampicillin. Protein expression was induced with 0.5 mM Isopropyl β-D-thiogalactoside (IPTG) when the OD600 nm of the culture reached 0.6–0.8 for 3 h at 37 ºC. All fusion proteins were purified from inclusion bodies under denaturing conditions using Ni-NTA agarose beads (Life Technologies) [39]. Purified proteins were concentrated with Amicon Ultra-15 MWCO 30,000 centrifugal filter units and stored in 8 M urea, 20 mM Tris, pH 8.0. Protein

concentration was determined using the Pierce Bicinchoninic Acid Protein Assay Kit (Thermo Fisher Scientific).

## Thioflavin T assays for amyloid assembly

The fluorescent amyloid reporter dye Thioflavin T was used to follow the kinetics of amyloid formation. Amyloid assembly was initiated by diluting proteins to a final protein concentration of 2.5 μM and urea concentration of 300 mM with 25 mM $NaH_2PO_4$, 150 mM NaCl, 40 μM ThT, 0.5 mM DTT, pH 7.4. Assays were conducted in triplicate, at 37°C in Costar black 96-well fluorescence plates (Corning) in a POLARstar Omega microplate reader (BMG Labtech) with fluorescence excitation at 440 nm and emission at 480 nm.

## Single molecule confocal spectroscopy

Proteins, either alone or in a mixture of one mCherry-tagged protein and one YPet-tagged protein, were diluted out of 8 M urea-containing buffer to a final concentration of 0.35–0.7 μM in 25 mM $NaH_2PO_4$, 150 mM NaCl, 0.5 mM DTT, pH 7.4. Samples were analysed at room temperature on a Zeiss Axio Observer microscope, in which two lasers (488 nm and 561 nm) were focused in solution using a 40×/1.2 NA water immersion objective. The fluorescence signal was collected and separated using a 565 nm dichroic mirror, with a 525/20 nm band pass filter for detection of YPet-tagged proteins and a 580 nm long pass filter for detection of mCherry-tagged proteins. For experiments with two fluorophores, the signals from the two channels were recorded simultaneously in 1 ms time bins. Photon-counting histograms were generated by binning signals of 10 photons. The fluorescence correlation analysis was performed as in Brown and Gambin [77].

## Confocal microscopy

Proteins, either alone or in mixtures, were dialysed at 5 μM from 8 M urea-containing buffer into 25 mM $NaH_2PO_4$, 150 mM NaCl, 0.5 mM DTT, pH 7.4. Samples (5 μL) were transferred to slides, covered with a coverslip and imaged on a Zeiss LSM 510 Meta Spectral Confocal Microscope with DAPI, GFP and Texas Red filters to visualise ThT, YPet and mCherry fluorescence, respectively. Images were analysed with ImageJ.

## Transmission electron microscopy

A droplet of protein solution (concentration 5 μM, 20 μL) was placed onto Parafilm and a carbon/formvar-coated copper grid (200 mesh, ProSciTech) floated on the surface for 1 minute, followed by removal of excess solution by wicking with filter paper. The grid was washed three times with filtered water, then stained with 2% uranyl acetate and imaged using a Philips CM120 microscope operating at 120 kV. Digital images were recorded using an EMSIS Veleta CCD camera and iTEM digital imaging system.

## Congo red staining

Ubiquitin-RHIM proteins (both WT and AAAA) were diluted to 0.3 mg/mL in 8 M urea-containing buffer and then were dialysed against an assembly buffer (25 mM $NaH_2PO_4$, 150 mM NaCl, 0.5 mM DTT, pH 7.4) overnight at room temperature. Congo red was added to 1 mL protein samples to a concentration of 2 μM in cuvettes. The absorbance spectrum of each protein solution was measured. Absorbance spectra for protein-containing samples were compared to Congo red only samples. Protein scattering effects were corrected by subtraction of absorbance across spectra of protein samples without Congo red.

## Sodium dodecyl sulfate agarose gel electrophoresis analysis

Samples of individual or combinations of proteins were diluted to 5 μM per protein in 8 M urea-containing buffer. A proportion of these samples were retained for use as a monomeric control. The remaining sample was dialysed against assembly buffer (25 mM $NaH_2PO_4$, 150 mM NaCl, 0.5 mM DTT, pH 7.4) overnight to allow for complete formation of protein assemblies. Glycerol was added to samples at a final concentration of 4% and bromophenol blue was added at a final concentration of 0.0008%. Samples were incubated with either SDS to a final concentration of 2% or MilliQ water for 10 min and then analysed as in [39]. Gels were imaged on a Chemi-Doc (BioRad) using 605/50 nm and 695/55 nm emission filters.

## Supporting information

**S1 Fig. Potential RHIMS in VZV strains.** (A) Amino acid sequence alignment of various VZV strains showing the ORF20 RHIM region and indicating the percentage of conservation and consensus sequence. Accession numbers of VZV sequences used: X04370.1, JF306641.2, JQ972913.1, DQ457052.1, KX262866.1, KU926322.1, DQ008355.1, DQ008354.1 The RHIM core is boxed (B) Amino acid sequence alignment of the RHIM identified in VZV ORF20 (Dumas) with potential RHIMs in the capsid triplex subunit 1 from other *Varicelloviruses*, Simian varicella virus (SVV), Cercopithecine herpesvirus 9 (CeHV-9), Bovine herpesvirus (BHV) -1 and -5, Bubaline herpesvirus 1 (BuHV-1), Equine herpesvirus (EHV) -, -4, -8 and -9, Pseudorabiesvirus (PRV), Feline herpesvirus -1 (FHV1), and Canine herpesvirus -1 (CaHV-1). (C) Immunoblot analysis of mock and VZV-infected HT-29s following with TNF (T; 30 ng/ml), BV-6 (S; 1 μM), z-VAD-fmk (V; 25 μM). * represents a non-specific band, arrows indicate protein size markers.
(TIF)

**S2 Fig. Infection by parental and RHIM mutant VZV.** (A) Average percentage of VZV-infected cells in the parental and VZV RHIM mutant virus HT-29 cultures at the beginning of each viability assay as determined by flow cytometry staining for the gE:gI glycoprotein complex. Error bars show standard error of the mean, from 4 independent replicates. (B) Viability of mock, VZV and VZV RHIM deleted (VZV-RHIMKO) virus infected HT-29s (72 h post-infection) following treatments with TNF (T; 30 ng/ml), BV-6 (S; 1 μM), z-VAD-fmk (V; 25 μM) and necrostatin-1 (Nec1; 30 μM) alone or in combination as indicated. Data was normalised to DMSO only control. Error bars show standard error of the mean, from 3 independent replicates and statistical significance was determined using a two-way ANOVA. C. Average percentage of VZV-infected cells in the parental and VZV RHIM deleted virus HT-29 cultures at the beginning of each viability assay as determined by flow cytometry staining for the gE:gI glycoprotein complex. Error bars show standard error of the mean, from 3 independent replicates.
(TIF)

**S3 Fig. ZBP1-expressing HT-29s with T+S+V retain the ability to undergo necroptosis.** ZBP1 expressing HT-29s were treated with DMSO (No treatment, NT) or with TNF (T; 30 ng/ml), BV-6 (S; 1 μM), z-VAD-fmk (V; 25 μM) for 18 h then cell viability measured using the Promega Cell Titre Glo2 assay.
(TIF)

**S4 Fig. Additional insight into the potential of ORF20 to interact with other RHIM-containing proteins.** Representative confocal fluorescence spectroscopy time profiles collected from (A) ORF20$_{1-114}$-mCherry and ORF20$_{1-114}$mut-mCherry with YPet-RIPK1$_{497-583}$, and (B)

YPet-ZBP1$_{170-355}$mutA or YPet-ZBP1$_{170-355}$mutB fusion proteins with ORF20$_{1-114}$-mCherry. Proteins were incubated alone or mixed in pairs under conditions that allow co-assembly. The proteins present in each mixture are indicated for each part of the figure. Inserts show detail of 1 s of the dual fluorescence recordings, at the time indicated by a star on the full time trace. (TIF)

**S5 Fig. Effect of RHIM mutation on the size of oligomers formed between ORF20 and ZBP1. (A)** Photon count histogram of fluorescence intensity detected in mCherry emission channel reflects oligomer particle size distribution over a 3 min period for ORF20$_{1-114}$-mCherry alone or in combination with YPet-ZBP1$_{170-355.}$ (B) Photon count histogram of fluorescence intensity detected in mCherry emission channel reflects oligomer particle size distribution over a 3 min period for ORF20$_{1-114}$mut-mCherry alone or in combination with YPet-ZBP1$_{170-355.}$ (C) Fluorescence correlation analysis reveals distribution of particles of different sizes, reflected by correlation coefficient (tau).
(TIF)

**S6 Fig. Schematic representations of the recombinant protein constructs used in this study.**
(TIF)

## Acknowledgments

The authors acknowledge the facilities and the scientific and technical assistance of Sydney Microscopy & Microanalysis Core Research Facility and The Molecular Biology Facility of the Bosch Institute at the University of Sydney. The authors would like to thank A/Prof. Susan McLennan (University of Sydney, NSW, Australia) for supplying the HT-29 cell line, Louise Cole, and the Bosch Institute Advanced Microscopy Facility at the University of Sydney for assistance and support with fluorescence microscopy, Professor Paul Kinchington (University of Pittsburg, PA, USA) for supplying the VZV BAC and antibodies, Elizabeth Hartland (Hudson Institute of Medical Research, Vic, Australia) for the GFP-ZBP1 expression construct and Dr James Murphy (Walter and Elizabeth Hall Institute, Vic, Australia), Prof Edward Mocarski (Emory University), and Ailis O' Carroll (University of NSW, Sydney, Australia) for helpful advice.

## Author Contributions

**Conceptualization:** Megan Steain, Barry Slobedman, Margaret Sunde, Allison Abendroth.

**Data curation:** Megan Steain, Max O. D. G. Baker.

**Formal analysis:** Megan Steain, Nirukshan Shanmugam, Margaret Sunde.

**Funding acquisition:** Megan Steain, Margaret Sunde, Allison Abendroth.

**Investigation:** Megan Steain, Max O. D. G. Baker, Chi L. L. Pham, Nirukshan Shanmugam, Brian P. McSharry, Selmir Avdic, Margaret Sunde.

**Methodology:** Megan Steain, Max O. D. G. Baker, Chi L. L. Pham, Nirukshan Shanmugam, Yann Gambin, Emma Sierecki, Brian P. McSharry, Margaret Sunde.

**Project administration:** Megan Steain, Margaret Sunde.

**Resources:** Yann Gambin, Emma Sierecki.

**Supervision:** Megan Steain, Chi L. L. Pham, Margaret Sunde, Allison Abendroth.

**Validation:** Megan Steain.

**Visualization:** Megan Steain.

**Writing – original draft:** Megan Steain, Max O. D. G. Baker, Margaret Sunde.

**Writing – review & editing:** Megan Steain, Max O. D. G. Baker, Chi L. L. Pham, Barry Slobedman, Margaret Sunde, Allison Abendroth.

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
