## [Decision Letter · Decision Letter 0]

8 Apr 2020

Dear Dr Steain,

Thank you very much for submitting your manuscript "Varicella zoster virus encodes a viral decoy RHIM to inhibit cell death" for consideration at PLOS Pathogens. As with all papers reviewed by the journal, your manuscript was reviewed by members of the editorial board and by several independent reviewers. In light of the reviews (below this email), we would like to consider the comments of the reviewers and invite the resubmission. This should be a significantly-revised version that takes into account all reviewers' comments with a cover letter describing all changes as well as reasons for not making requested modifications.

We cannot make any decision about publication until we have seen the revised manuscript and your response to the reviewers' comments. Your revised manuscript is also likely to be sent to reviewers for further evaluation.

Sincerely,

Edward Mocarski

Associate Editor

PLOS Pathogens

Shou-Jiang Gao

Section Editor

PLOS Pathogens

Kasturi Haldar

Editor-in-Chief

PLOS Pathogens

orcid.org/0000-0001-5065-158X

Michael Malim

Editor-in-Chief

PLOS Pathogens

orcid.org/0000-0002-7699-2064

Reviewer's Responses to Questions

**Part I - Summary**

Reviewer #1: Programmed cell death is a process in which the cells activate intracellular death pathways to terminate themselves in a systematic way in response to a wide variety of stimuli. Viruses must replicate within host cells and modulate host cell programmed death pathway. Herpesviruses are known to encode a number of inhibitors of host cell death, including Rip Homotypic Interaction Motif (RHIM)-containing proteins. In this article, the authors analyze the interaction between the programmed cell death and Varicella zoster virus (VZV), which is a member of the alphaherpesvirus subfamily and is responsible for causing chickenpox and shingles. The authors identified a novel viral RHIM in the VZV capsid triplex protein open reading frame (ORF) 20 that acts as a host cell death inhibitor. The ORF20 RHIM is capable of forming fibrillar amyloid complexes similar to that of M45 from murine cytomegalovirus. The ORF20 RHIM is not responsible for inhibition of TNF-mediated necroptosis, but is responsible for the ZBP1 pathway.

The article is well written, and these results suggested the importance of RHIM domain in VZV ORF20 for preventing ZBP1-driven apoptosis during VZV infection. The authors also propose that ORF20 RHIM sequesters ZBP1 into amyloid to prevent downstream signaling. This article will be of broad interest in the readership. However, the reviewer suggests following comments to improve this article.

Reviewer #2: This paper reports the surprising and rather unexpected finding that a capsid protein encoded by VZV encodes a RHIM type protein that is not involved in the blockade of necroptosis but is involved in the blocking mediation of the DNA/RNA sensor DAI/ ZBP-1 in triggering apoptosis. The data convincingly shows that the ORF20 protein, one of the triplex capsid proteins, has a predicted RHIM motif that, when mutated, doe not result in a virus that is now sensitive to necroptosis. They conclude ORF20 is not involved ( this leaves the question of what other protein or mechanism is responsible for blocking necroptosis, but I do not think this is needed for this paper). They Show that ORF20 does not interacts with the RIPK3 proteins involved in necroptosis, but rather with DAI in forming fibrils. This blocks DAI induced apoptosis. There are some nice studies on the fibrils formed and size/shape etc. Overall this is an important contribution that is first, not expected and second, supported by good data. The discussion is nice and succinct

Reviewer #3: The manuscript by Steain et al. identified a putative RHIM in the VSV capsid triplex protein ORF20 of VZV. Viral RHIM proteins have been identified in other herpesviruses, including M45 of CMV and ICP6/10 of HSV1/2. In support of ORF20 encoding a RHIM, expression of RHIM-fusion constructs formed amyloid-like complexes; however, other data presented does not support that ORF20 functions as a “bona fide” RHIM. 1) Binding to ZBP1 and RIPK3 was not dependent on the core residues of the RHIM that have been previously shown to be essential for TRIF, RIPK1, RIPK3, M45, ICP6/10 to interact with one another. 2) Necroptosis induction requires RHIM-interactions and ORF20 does not either promote or prevent necroptosis. Overall, there is indeed remarkable similarity of ORF20 to known cellular viral; however, the data presented does not provide sufficient evidence to support that ORF20 functions as a RHIM antagonist (or agonist) or data that accounts for such an unanticipated non-classical role for a VZV “RHIM.”

**Part II – Major Issues: Key Experiments Required for Acceptance**

Reviewer #1: 1. In this article, the authors described the importance of ORF20 for protection from programed cell death and these results clearly show the interaction between the ORF20 RHIM domain and ZBP1. However, the data in infected cells is lacking. The authors should show the importance of the RHIM domain in ORF20 for viral growth in normal cells such as MRC-5 cell.

2．As the authors described, ZBP1 is considered to be function in the cytoplasm, although capsid proteins of herpesvirus localize in the nucleus mainly. The authors should analyze the colocalization between ZBP1 or RIPK3 and ORF20 in infected cells.

3． In this article, the authors argued that the ORF20 RIHM inhibits ZBP1-mediated apoptosis in infected cells but not necroptosis. However, the results that VZV ORF20 interacts with ZBP1 and RIP3 could not explain this difference. Can the ORF20 inhibit necroptosis in transient system?

Data in Fig. 4 should be quantified and analyzed statistically.

Reviewer #2: The VZV mutants need at least growth curve analyses and comparison on ARPE19 cells

Also did the authors look at the larger ORF20 deletion mutant for growth and blockade of necroptosis induction? It is possible that just because the AAAA mutant stopped RIP3K interaction in MVMV and HSV, it might not be enough here. This would be more severe than the single four AAAA substitution, which might still interact with RIPK to stop necroptosis through the flanking homology (it can still interact with DAI/ZBP-1….).

Reviewer #3: The text that accompanies Figure 1 suggests that VZV sensitized HT-29 cells to TNF-induced apoptosis. While statistics are shown, the text does not accurately reflect the modest sensitization. Here a mechanism would be of interest. Given there is no difference in apoptosis in T+S, does VSV affect cIAP levels? Additionally, the necroptosis inhibition by VSV was modest especially compared to Nec-1 inhibition or inhibition by the ICP6 (Guo et al, 2015) in HT-29 cells.

In Figure 2, it is unclear how VZV infection conferred resistance to necroptosis in neighboring uninfected cells. Does ORF20 “RHIM” mutation influence VSV replication/infectivity?

Figure 4 indicates that ZBP1 expressing HT29 cells form fewer infectious centers than parental HT29 cells and that this could be modulated by caspase inhibition but not the necroptosis inhibitor NSA. Western blots or IF for activated caspases should be incorporated to make certain zVAD rescue is on target. Additionally, ZBP1-induced apoptosis occurs via caspase 8 activation via RIPK3-RIPK1 during Influenza infection. Does Nec-1 rescue? Does KO or knockdown of RIPK3 and RIPK1 rescue? FADD dominant negative? ZBP1-expressing HT-29 cells are very sensitive to Influenza (PR8) and HSV1-ICP6mutRHIM-induced necroptosis. Are VSV-infected ZBP1 HT29 cells resistant to ZBP1-induced necroptosis by either Influenza or mutant HSV1?

The binding assays presented in Figure 6 demonstrate binding of ORF20 to ZBP1 and RIPK3 independent of the ORF20 RHIM core. Inclusion of RIPK1 and TRIF would be informative. Mutation of the RHIM domain in RIPK3 and ZBP1 should be shown. The ORF20 RHIM-KO similar to Figure 4 or other mutation that eliminates binding should be identified.

**Part III – Minor Issues: Editorial and Data Presentation Modifications**

Reviewer #1: 1. In Fig. 2B, please show the results of mock with and without TSV.

2. In the legend of Fig. 3, please indicate time after infection.

3. In Fig. 4 and 5, please indicate a strain name instead of Parent in the label.

4. Please indicate the protein detected for infection in Fig. S1C.

Reviewer #2: 1. I have a few questions and clarification issues. The HSV capsid protein is not included in figure 1 alignment. Is there any sign of a RHIM like motif in it? Secondly, is it detectable in the Beta and Gamma herpesvirus equivalents

2. Fig1c. Cell images appear to be different sizes and magnifications and need a size bar

3. The VZV mutants need at least growth curve analyses and comparison on ARPE19 cells

4. Line 190 how long past infection were the studies done? And the level of necroptosis seems rather weak, and the effect of inhibition is modest. Also did the authors look at the larger ORF20 deletion mutant for growth and blockade of necroptosis induction? It is possible that just because the AAAA mutant stopped RIP3K interaction in MVMV and HSV, it might not be enough here. This would be more severe than the single four AAAA substitution, which might still interact with RIPK to stop necroptosis through the flanking homology (it can still interact with DAI/ZBP-1….).

5. Figure 2A could be a lot more convincing. The cells showing MKLK and much less dense and the red signal is barely visible. Indeed, the correlation with VZV and uninfected cells could be improved by showing flow plots and the red and green signals in black and white, separate channels rather than overlap. Why is the MKLK only being activated in 20% of cells? this seems rather low, and others using HRT1080s have gotten much higher levels.

6. S1C showing P-MKLK should go into the main figures. Also why is MKLK missing from two lanes in total assessment levels?

7. Lin 265 What are the endogenous levels of ZBP1 in the cells transfected ? how much higher levels are they needing to see the effect?

8. Given that the mutant ORF20 proteins still form a complex with ZBP-1 even though it is rather atypical, why does it still not inhibit apoptosis? A bit more discussion is needed

Reviewer #3: Acknowledging, the challenge of transducing HT29 cells with a ORF20 expression construct, does ORF20 expression in other necroptosis-sensitive cells such as murine L929 or SVEC4-10 or ZBP1/RIPK3 transduced 293T or HeLa suppress necroptosis/necroptosis?

PLOS authors have the option to publish the peer review history of their article (what does this mean?). If published, this will include your full peer review and any attached files.

Reviewer #1: No

Reviewer #2: No

Reviewer #3: No
---

## [Editor Report · Decision Letter 1]

28 May 2020

Dear Dr Steain,

We are pleased to inform you that your manuscript 'Varicella zoster virus encodes a viral decoy RHIM to inhibit cell death' has been provisionally accepted for publication in PLOS Pathogens.

Best regards,

Edward Mocarski

Associate Editor

PLOS Pathogens

Shou-Jiang Gao

Section Editor

PLOS Pathogens

Kasturi Haldar

Editor-in-Chief

PLOS Pathogens

orcid.org/0000-0001-5065-158X

Michael Malim

Editor-in-Chief

PLOS Pathogens

orcid.org/0000-0002-7699-2064

The manuscript was revised in response to reviewers' comments within acceptable limits that have been placed on all research by the COVID-19 pandemic.

The edits, additional data and corrections all dramatically improve the manuscript.
---

## [Editor Report · Acceptance letter]

6 Jul 2020

Dear Dr Steain,

We are delighted to inform you that your manuscript, "Varicella zoster virus encodes a viral decoy RHIM to inhibit cell death," has been formally accepted for publication in PLOS Pathogens.

Best regards,

Kasturi Haldar

Editor-in-Chief

PLOS Pathogens

orcid.org/0000-0001-5065-158X

Michael Malim

Editor-in-Chief

PLOS Pathogens

orcid.org/0000-0002-7699-2064